# Using Remote Sensing and Climate Data to Map the Extent and Severity of Balsam Woolly Adelgid Infestation in Northern Utah, USA

Michael J. Campbell [1,*], Justin P. Williams [2] and Erin M. Berryman [3]

1   Department of Geography, University of Utah, 260 South Central Campus Drive,
    Salt Lake City, UT 84112, USA
2   Forest Health Protection, USDA Forest Service, 4746 South 1900 East, Ogden, UT 84403, USA;
    justin.williams3@usda.gov
3   Rocky Mountain Research Station, USDA Forest Service, 240 West Prospect Road,
    Fort Collins, CO 80526, USA; erin.berryman@usda.gov
*   Correspondence: mickey.campbell@geog.utah.edu

**Abstract:** Balsam woolly adelgid (Hemiptera: *Adelges picea* Ratzeburg; BWA) is a nonnative, invasive insect that has infested fir trees in the US for over a century, yet robust methods for mapping BWA have remained elusive. We compare three approaches to mapping BWA in the subalpine fir forests of northern Utah, the forefront of BWA spread in the western US: (1) using moderate-resolution, multispectral satellite imagery; (2) using terrain and climate data; and (3) using a combination of imagery, terrain, and climate data. While the spectral data successfully detected forest degradation, they failed to distinguish between causal agents of change ($R^2_{mean}$ = 0.482; $RMSE_{mean}$ = 0.112). Terrain and climate data identified landscape conditions that promote BWA infestation but lacked the ability to characterize local-scale tree damage ($R^2_{mean}$ = 0.746; $RMSE_{mean}$ = 0.078). By combining spectral, terrain, and climate data, we present a repeatable approach for accurately mapping infestation severity that captures both regional abiotic drivers and the local damage signals of BWA ($R^2_{mean}$ = 0.836; $RMSE_{mean}$ = 0.065). Highly infested areas featured increased visible and shortwave infrared reflectance over time in the spectral data. The terrain bore little influence on severity, but climate variables indicated that warmer areas are more prone to severe infestation. This research study presents an analytical framework upon which future BWA monitoring efforts can be built.

**Keywords:** balsam woolly adelgid; *Adelges piceae*; subalpine fir; *Abies lasiocarpa*; invasive species; forest entomology; forest health; remote sensing; climate; random forests

## 1. Introduction

The balsam woolly adelgid (Hemiptera: *Adelges picea* Ratzeburg; hereafter BWA) is an introduced insect that affects North American true firs (*Abies* spp.), causing tree damage and mortality among all age classes [1]. Native to Europe, BWA entered North America in the early 1900s via separate introductions on the east [2] and west coasts [3]. In the western United States, BWA has slowly invaded true fir forests from northern California [3] to Oregon [4], Washington [5], Idaho in 1983 [6], Montana in 2007 [7], and as of 2017, Utah [7]. The relative threat of BWA to true fir forests varies geographically, as the severity of impacts can vary by host species, site, stand age and conditions, and local climate [8–11]. In Utah, the threat posed by BWA is high given the abundance and ecological value of a particularly susceptible host, subalpine fir (*Abies lasiocarpa* (Hook.) Nutt.). Subalpine fir is the sixth-most abundant species in the state by basal area, according to FIA data [12,13]. It is a late-seral, shade-tolerant species often found dominating poor sites at high altitudes, occupying a unique and important niche among Utah's forests as a wildlife habitat and acting as a critical carbon sink. Accordingly, the novel invasion of BWA into the state has the potential for devastating effects. A second true

fir species in Utah, white fir (*Abies concolor* (Lindley ex Hildebrand) Gordon), can be infested by BWA but is far less susceptible to severe impacts and mortality [14].

The life history, dispersal, and cryptic nature of BWA create unique challenges for the detection and management of this pest. BWAs are microscopic (<1 mm) in size and parthenogenic, producing between two and four generations per growing season in the northwest USA [15]. The mobile stage in BWA's life cycle is the first instar "crawler" when BWAs first hatch from the egg and either settle on the same tree as their parent or are passively dispersed via wind or phoretic movement on birds [16]. After selecting a feeding site on the host and inserting their stylet into host tissues, they become immobile. In the final generation of the growing season, settled first instar BWAs overwinter and begin the following year's population. Cold winter temperatures ($<-20\ °C$) can significantly reduce the overwintering population; however, refuge under snowpack insulates BWAs at the lower portions of the bole from lethal temperatures, reducing overwinter mortality [10]. As BWAs progress through three instars toward adulthood they produce the protective wax-like "wool", which surrounds their body and is the first sign of infestation. In the beginning stages of infestation, BWA and wool can be difficult to identify, even in detailed, field-based inventories. In addition, another woolly adelgid in this region, *Pineus abietinus* Underwood & Balch, also infests true firs and appears identical to BWA [17,18]. Identifying the species must be carried out via a morphological examination of a slide-mounted specimen or via DNA extraction [19]. As the BWA population establishes and grows, a symptom unique to BWA, gouting (abnormal swelling of branch nodes), can be used to identify and confirm this damage agent [20,21]. Once established in a stand, BWA populations may fluctuate year-to-year but will persist indefinitely so long as live hosts are available [11]. Management strategies for BWA are limited to promoting non-host tree species via silvicultural treatment or planting. Attempts at slowing BWA population growth and damage due to insecticides are impractical at landscape scales. Silvicultural management options include increasing stand vigor and the removal of highly infested trees; however, evidence of their effectiveness is lacking.

Remote sensing is a critical tool for mapping the spatially and temporally explicit extent and severity of insect-induced disturbances in forest ecosystems [22–25]. By exploiting changes in spectral data over time (magnitude, timing, spatial patterns, etc.), we can identify areas where significant changes to forest canopy conditions have occurred. Ideally, these changes can be linked to one or more disturbance agents via a comparison to spatially coincident reference data, provided that a statistically robust relationship can be formed between the spectral change and the agent(s). This general approach has been demonstrated to be successful in a broad range of forest ecosystems [26–31]. However, the bulk of this research study has focused on bark beetles and canopy defoliators [25]. Unlike these insects, which tend to produce relatively rapid and spatially clustered changes in forest canopy conditions and associated spectral responses, the complex spatiotemporal nature of BWA infestation has resulted in a relative paucity of remote-sensing-focused BWA studies. For example, tree health impacts from BWA infestation can unfold over the course of several years, and although mortality may be an eventual outcome, it is not guaranteed [1]. Furthermore, the spatially diffuse nature of infestation attributed to wind-driven insect dispersal increases the likelihood that a diverse range of infestation severities may be present even within a single satellite image pixel of moderate spatial resolution (e.g., 30 m), obscuring the stand-level spectral signal of infestation [16,32]. Lastly, BWA is often only one among an array of agents acting in concert to damage a tree. For example, a tree can be weakened by BWA, but a subsequent infestation of bark beetles may be the dominant cause of mortality [33]. Thus, attributing tree damage, both on the ground and from above, specifically and uniquely to BWA can be challenging.

Despite these limitations, a few studies have been successful at applying remote sensing to the study of BWA infestation. Franklin et al. [34] demonstrated the novel capacity of classifying BWA infestation severity at the level of the individual tree; however, their approach relies on a relatively rare combination of high-spatial-resolution (0.5–1.0 m)

and spectral resolution (288 bands) image data, limiting broad applicability. For example, satellite-based sensors with comparable spectral resolution, such as the DLR Earth Sensing Imaging Spectrometer (DESIS) and Hyperspectral Precursor of the Application Mission (PRISMA), have spatial resolutions of 30 m [35,36]. Cook et al. [32] used reflectance spectra from a field spectrometer to characterize branch-level BWA infestation. However, they only characterized infestation in a binary fashion (infested vs. non-infested). From an ecological and management perspective, it would be more useful to be able to distinguish relative degrees of infestation (e.g., low, moderate, and severe) in order to understand relative impacts. Furthermore, while they did convolve field spectra to simulate satellite imagery, they note that using real remote sensing data would be challenging due to spectral mixing at the pixel level and the presence of multiple stressors on host trees obscuring the relatively subtle BWA-specific signal. To our knowledge, Hutten [37] has been the only one carrying out an attempt at using a time series of remote sensing data to map BWA infestation at the stand level, finding that low-level change in the normalized burn ratio over time had statistically significant relationships with the presence of BWA infestation symptoms from field and aerial surveys. As with Cook et al. [32], however, Hutten's [37] analysis did not quantify infestation severity on a continuous scale; instead, they merely distinguished between the presence and absence of BWA. Although these studies demonstrate promise, there remains a need to better understand the capacity for mapping BWA infestation on a continuous scale using widely available remote sensing time series data. Being able to map relative degrees of infestation provides land managers with a greater ability to prioritize mitigation efforts and understand potential future spread.

If remote sensing alone is insufficient for mapping BWA to a desirable degree of accuracy and precision, then perhaps additional geospatial data can augment the analysis. BWA damage has been shown to be strongly temperature- and terrain-dependent [10,37–42]. Although many have noted this dependency, few have exploited it for mapping purposes. Hrinkevich et al. [39] compared plot-level BWA infestation severity to a suite of climate variables derived from PRISM data [43], finding summer and autumn temperature-related variables to be particularly important in predicting severity. This study provides evidence of the potential benefit of utilizing spatial data representing abiotic environmental factors, but the predictive power of their models was fairly low ($R^2 = 0.24$), and their results were generated at a relatively coarse spatial resolution (4 km). Thus, there remains a need to map infestation severity at a resolution and predictive accuracy that are of greater use to forest managers who require more spatially precise data to drive stand-level management decisions.

Remote sensing data can identify changes in forest conditions over time, although they often pose challenges with respect to distinguishing between causal agents with similar spectral signals [26,44]. Terrain and climate data can map susceptibility to insect infestation, particularly among species like BWA who have shown a strong climatic dependency, although susceptibility alone does not directly translate to certain infestations or their relative severity [39]. Increasingly, there is a recognition that the combined use of remote sensing and spatially explicit representations of abiotic variables may exceed the capabilities of each data type used in isolation [25,28,45]. Given BWA's relatively subtle damage symptoms and demonstrated climatic dependency, this study aims to leverage the individual and combined use of spectral, terrain, and climate data to gain a robust understanding of the strengths and limitations of mapping BWA infestation severity.

The objectives of this study were to carry out the following:

1.  Develop an accurate map of current BWA infestation severity for use by land managers, focusing on the relatively recent invasion of northern Utah;
2.  Compare remote sensing-driven and terrain-/climate-driven approaches to mapping BWA infestation severity using a field-validated quantitative measure of stand-level severity;
3.  Introduce a new approach for mapping BWA infestation severity that leverages individual strengths and overcomes the individual weaknesses of remote sensing and terrain/climate data;

4. Produce a quantitative accounting of landscape-level environmental drivers and identify key geospatial predictors of BWA infestation severity.

## 2. Materials and Methods

### 2.1. Study Area

This study took place within the Uinta–Wasatch–Cache National Forest (UWCNF) in northern Utah, USA (Figure 1). This area was chosen because BWA infestation was first detected relatively recently (2017), and infestation severity varies widely throughout the forest's extent. Centered at approximately 40°51′35.62″ N and 111°21′15.14″ W, UWCNF encompasses 11,788 km² of land within its administrative boundary, containing multiple ownerships. Most of the forest is within the Wasatch and Uinta Mountains, with the former running north–south on the west side of the forest and the latter running east–west to the east, spanning a wide range of elevations from approximately 1300 m to 3900 m. Likewise, the climate varies widely; 30-year average annual temperatures range from approximately −3 °C to 12 °C, and annual precipitation totals range from about 280 mm to 1680 mm. From lower, hotter, and drier elevations to higher, cooler, and wetter portions of the forest, the most common tree species range from Utah juniper (*Juniperus osteosperma* (Torr.) Little) to Gambel oak (*Quercus gambelii* Nutt.), Douglas fir (*Pseudotsuga menziesii* (Mirb.) Franco), lodgepole pine (*Pinus contorta* Douglas ex Loudon), quaking aspen (*Populus tremuloides* Michx.), subalpine fir (*Abies lasiocarpa* (Hook.) Nutt.), and Engelmann spruce (*Picea engelmannii* Parry ex Engelm.). Species such as two-needle piñon (*Pinus edulis* Engelm.), single-leaf piñon (*Pinus monophylla* Torr. & Frém.), bigtooth maple (*Acer grandidentatum* Nutt.), ponderosa pine (*Pinus ponderosa* Lawson & C. Lawson), white fir (*Abies concolor* (Gord. & Glend.) Lindl. ex Hildebr.), limber pine (*Pinus flexilis* James), and a variety of riparian tree species (e.g., *Populus* spp. and *Salix* spp.) are also present throughout the forest, but they are less common. The primary host species for BWA in this region is subalpine fir, and it is found throughout UWCNF.

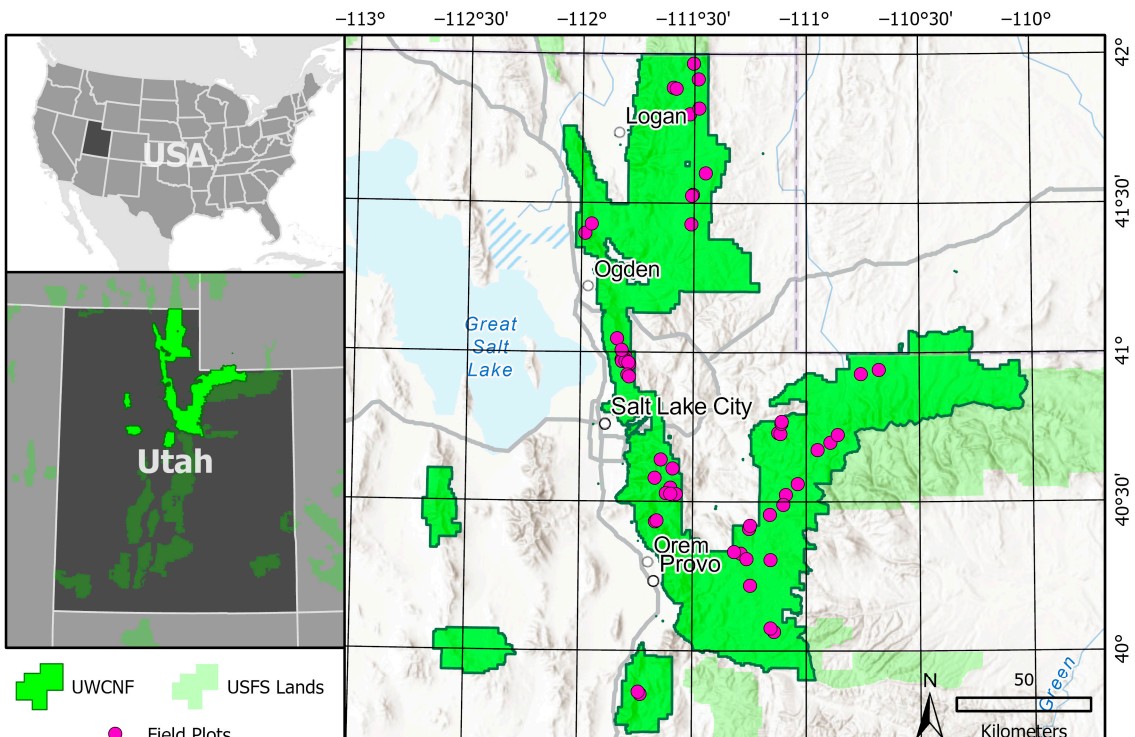

**Figure 1.** Study area map of Uinta–Wasatch Cache National Forest (UWCNF) with field plot locations placed within the context of other USDA Forest Service (USFS) lands within Utah, USA. Basemap credit: Esri.

## 2.2. Field Data

To train and test our predictive mapping models, we collected data from 58 plots during 2021 and 2022. BWA population densities peak in the autumn, so data were collected between the months of September and October in both years. In year 1 (2021), plot locations were determined using a conditioned Latin hypercube sampling strategy [46] aimed at capturing variability in terrain (e.g., elevation and aspect), spectral information (e.g., Landsat NDVI change over time), and expert knowledge-informed manually digitized delineations of known infestation levels. In year 2 (2022), plot locations were opportunistically selected based on preliminary modeling results that used year 1's data to improve areas that were likely over- or under-estimated with respect to infestation severity. Throughout both years, plots were placed at least 500 m apart in areas with a high proportional abundance of host tree species, and efforts were made to distribute the plots spatially across the extent of UWCNF (although plots tended to be somewhat clustered for efficiency).

We developed a field protocol aimed at capturing tree species composition and BWA infestation severity, while accounting for other damage agents (e.g., bark beetles and pathogens), to minimize confusion in the modeling process. This protocol was informed by Hrinkevich et al. [47], including the use of several of their rating systems for evaluating the severity of specific infestation symptoms. Plots were circular with a fixed radius of 15 m, and the goal was to approximately and spatially match the 30 m resolution remote sensing data and other predictor data. Plot center locations were recorded using a Trimble R1 GNSS receiver using $\geq$100-point position averaging, resulting in an average positional accuracy of 1.04 m (SD = 0.49 m). Within each plot, every standing tree (defined as $\geq$45° above the ground surface) with a stem diameter greater than 5 cm was evaluated for each metric in Table 1. All data were recorded in Esri Survey123.

**Table 1.** Tree-level metrics were observed and recorded for every eligible tree within our field plots.

| Metric | Description |
| --- | --- |
| Species | Tree species |
| Status | Categorical indicator of the tree's vitality:<br><br>• Live (any live foliage present);<br>• New dead (no live foliage, but needles still present);<br>• Old dead (no live foliage, no needles present).<br><br>Note that for old dead trees, only species and DBH were recorded. |
| DBH | Diameter at breast height in cm |
| Wool Density | Categorical measure of BWA wool density on the lower 6 ft (1.83 m) of the tree bole, measured in wools per ft$^2$ (929 cm$^2$):<br><br>• 0: 0 wool/ft$^2$;<br>• 1: >0–10 wool/ft$^2$;<br>• 2: >10–100 wool/ft$^2$;<br>• 3: >100 wool/ft$^2$. |
| Gout Severity | Categorical measure of the degree to which branches and twigs have developed gouts, as approximated based on their noticeability (Figure 2):<br><br>• 0: None;<br>• 1: Light (barely noticeable, even from close proximity, Figure 2A);<br>• 2: Moderate (easily noticeable from close proximity, Figure 2B);<br>• 3: Severe (easily noticeable from a distance, Figure 2C). |
| Crown Deformities | Count of the number of crown deformities observed in the top few meters of the tree. Candidate deformities include stunted growth in the terminal branch (leader), stunted growth in the lateral branches, and a top curl (Figure 3). Values range from 0 to 3. |
| Dead Top | Binary (0: no; 1: yes) indicator of whether or not the uppermost portion of the tree's crown is dead. |

Table 1. *Cont.*

| Metric | Description |
| --- | --- |
| Branch Dieback | Proportion of the tree's retained foliage that is no longer photosynthetically active (yellow, red, or brown), in percentage classes at a 10% interval (e.g., 0%, 10%, . . . , 100%). |
| Other Damage Agents | Indicator of evidence of any non-BWA agents that have caused damage to the tree's health, including (but not limited to) fir broom rust, bark beetles, twig beetles, pathogens, mechanical damage, and frost crack. |
| BWA Damage Score (*BDS*) | Integrated qualitative indicator of our perception of the degree of damage caused by BWA:<br><br>• 0: None (BWA is not present);<br>• 1: Light (BWA is present but the tree remains largely unaffected);<br>• 2: Moderate (BWA is present and is having a moderate impact on tree health);<br>• 3: Severe (BWA is present and is having a severe impact on tree health).<br><br>All newly dead trees were attributed with either a *BDS* of 3 (if we interpreted BWA as the primary mortality agent), an *ODS* of 3 (if we interpreted other sources as the primary mortality agent(s)), or both. |
| Other Agent Damage Score (*ODS*) | Integrated qualitative indicator of our perception of the degree of damage caused by other agents:<br><br>• 0: None (other agents not present);<br>• 1: Light (other agents are present but the tree remains largely unaffected);<br>• 2: Moderate (other agents are present and are having a moderate impact on tree health);<br>• 3: Severe (other agents are present and are having a severe impact on tree health).<br><br>All newly dead trees were attributed with either a *BDS* of 3 (if we interpreted BWA as the primary mortality agent), an *ODS* of 3 (if we interpreted other sources as the primary mortality agent(s)), or both. |

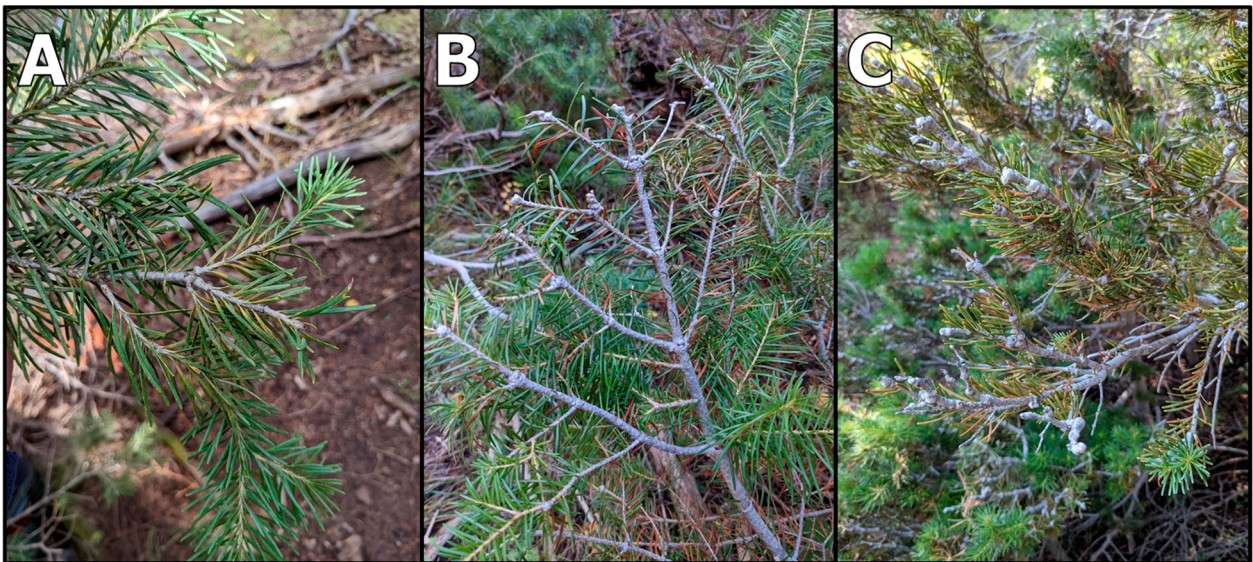

**Figure 2.** Example photographs of gout severity classes: (**A**) light, (**B**) moderate, and (**C**) severe.

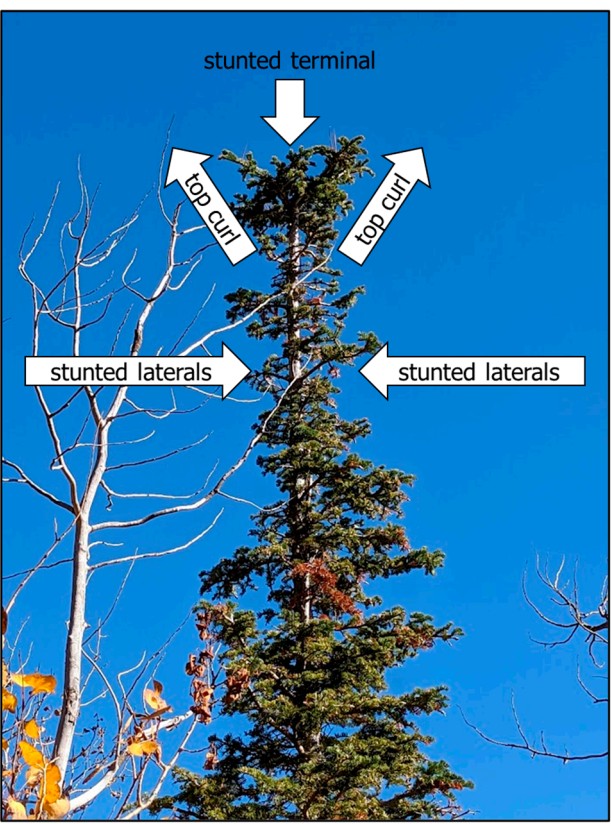

**Figure 3.** Example of a tree that features three crown deformities, as defined in our field protocol.

To derive plot-level estimates of infestation severity, we first normalized all tree-level infestation indicator metrics (wool density, gout severity, crown deformities, dead top, and branch dieback) to a consistent 0–1 scale. To distinguish tree-level damage caused by BWA from that caused by other agents, we calculated the total damage score (*TDS*) for each tree, which is equal to the sum of the BWA damage score (*BDS*) and other agent damage score (*ODS*) (Table 1; Equation (1)). We then calculated the proportional BWA damage score (*pBDS*) and proportional other agent damage score (*pODS*) by dividing *BDS* and *ODS* by *TDS*, both of which ranged from 0 to 1 (Equations (2) and (3)).

$$TDS = BDS + ODS \tag{1}$$

$$pBDS = BDS/TDS \tag{2}$$

$$pODS = ODS/TDS \tag{3}$$

To link tree-level normalized infestation indicator metrics to tree-level proportional damage agent scores, we multiplied *pBDS* by each of the normalized infestation indicator metrics, resulting in a series of metrics weighted by their proportional attribution to BWA. For example, if normalized crown deformities for a particular tree were 1 and *pBDS* was 0.5, the BWA-weighted normalized crown deformities would be 0.5, suggesting that crown deformities were driven half by BWA and half by other agents. Lastly, we aggregated tree-level severity to plot-level severity by calculating the mean of all BWA-weighted normalized infestation indicator metrics for every subalpine fir tree in each plot.

### 2.3. Remote Sensing Data

We used a 10-year time series of Landsat 8 OLI and Landsat 9 OLI-2 data to link infestation severity to spectral change over time. These moderate-resolution (30 m) multi-

spectral (7 bands) sensors provide spectral reflectance data from the visible to the shortwave infrared portion of the electromagnetic spectrum with an 8- (post-Landsat 9) or 16-day (pre-Landsat 9) return interval. Please note that all references to "spectral data" used in this study henceforth refer to Landsat multispectral image data. As stated earlier, the infestation was first detected within the study area in 2017; however, it is likely, given the time difference between the initial infestation and the severity of infestation symptoms at the time of detection, that the initial infestation occurred prior to 2017. Additionally, we wanted to avoid spectral differences between OLI/OLI-2 and prior Landsat sensors (TM and ETM+) to ensure a consistent time series. Thus, we analyzed data from 2013 (the first year of Landsat 8 OLI availability) to 2022 (the most recent year of data available at the time of analysis). Annual image composites were generated in Google Earth Engine [48] by calculating per-band pixel-level median surface reflectance from snow, cloud, cloud shadow, and spectral saturation-masked USGS Landsat 8 Level 2, Collection 2, Tier 1 data within a temporal window from June 1st to September 30th for each year in the time series. This masking was carried out using the QA bands associated with the image data. In addition to the raw spectral reflectance data for each of the seven OLI/OLI-2 bands, a suite of vegetation indices was generated for each of the 10 years of data (Table 2).

**Table 2.** Vegetation indices generated from Landsat 8 OLI and Landsat 9 OLI-2 image data for use in the spectral time series analysis, after [49].

| Index | Abbreviation | Formula | Source |
|---|---|---|---|
| Normalized Difference Vegetation Index | NDVI | $\frac{(NIR - Red)}{(NIR + Red)}$ | [50] |
| Enhanced Vegetation Index | EVI | $2.5 \times \frac{(NIR - Red)}{(NIR + 6 \times Red - 7.5 \times Blue + 1)}$ | [51] |
| Near Infrared Reflectance of Vegetation | NIR$_V$ | $NIR \times NDVI$ | [52] |
| Soil Adjusted Vegetation Index | SAVI | $1.5 \times \frac{(NIR - Red)}{(NIR + Red + 0.5)}$ | [53] |
| Modified Soil Adjusted Vegetation Index | MSAVI | $\frac{\left(2 \times NIR + 1 - \sqrt{(2 \times NIR + 1)^2 - 8 \times (NIR - Red)}\right)}{2}$ | [54] |
| Normalized Difference Moisture Index | NDMI | $\frac{(NIR - SWIR_1)}{(NIR + SWIR_1)}$ | [55] |
| Normalized Burn Ratio | NBR | $\frac{(NIR - SWIR_2)}{(NIR + SWIR_2)}$ | [56] |
| Normalized Burn Ratio 2 | NBR2 | $\frac{(SWIR_1 - SWIR_2)}{(SWIR_1 + SWIR_2)}$ | [57] |
| Tasseled Cap Brightness | TCB | $0.3029 \times Blue + 0.2786 \times Green + 0.4733 \times Red$ $+0.5599 \times NIR + 0.5080 \times SWIR_1 + 0.1872 \times SWIR_2$ | [58] |
| Tasseled Cap Greenness | TCG | $-0.2941 \times Blue - 0.2430 \times Green - 0.5424 \times Red$ $+0.7276 \times NIR + 0.0713 \times SWIR_1 - 0.1608 \times SWIR_2$ | [58] |
| Tasseled Cap Wetness | TCW | $0.1511 \times Blue + 0.1973 \times Green + 0.3283 \times Red$ $+0.3407 \times NIR - 0.7117 \times SWIR_1 - 0.4559 \times SWIR_2$ | [58] |

Unlike other agents of change, such as fire; timber harvesting; or even other insects such as bark beetles, which tend to produce rapid changes in vegetation condition, the change caused by BWA infestation takes place over multi-year timescales. As a result, the spectral change caused by BWA infestation is relatively subtle and requires tailored analytical strategies capable of quantifying gradual spectral change over longer time periods. In this study, we relied on two pieces of temporal information as the basis of modeling BWA using spectral data: (1) spectral conditions at the start of our time series (2013) and (2) the slope of a regression line representing the general trend of change over the 10 annual time steps of spectral data. We generated these two metrics for each of our 18 spectral variables (7 raw bands + 11 vegetation indices), resulting in 36 candidate spectral predictors for modeling infestation severity. We explored the use of existing tools such as LandTrendr [59] and Continuous Change Detection and Classification [60], both of which employ a comparable analytical framework for assessing spectral change over time. However, these tools are both computationally intensive, require careful parameterization,

and are better-suited for characterizing temporally and spectrally stark land cover changes than subtle changes [61].

### 2.4. Terrain and Climate Data

In order to understand the potential influence of terrain on BWA infestation, we derived a suite of topographic metrics from 30 m resolution digital elevation models (DEM) that were acquired from the USGS 3D Elevation Program (Table 3). In total, there were 38 terrain predictor variables.

**Table 3.** Terrain variables derived from DEM data used in the prediction of BWA infestation severity.

| Abbreviation | Description |
| --- | --- |
| ASPECT_COS | Cosine of the direction of steepest decline, representing north–south-ness |
| ASPECT_SIN | Sine of the direction of steepest terrain decline, representing east–west-ness |
| CURV_PLAN | Curvature of the terrain in the perpendicular direction to the slope [62] |
| CURV_PROF | Curvature in the terrain in the parallel direction to the slope [62] |
| ELEV | Raw elevation data from DEM |
| HLI | Measure of solar radiation that incorporates slope, aspect, and latitude [63] |
| IMI | Wetness measure that incorporates accumulation of water flow, local curvature, and exposure to solar radiation [64] |
| SD_x | Standard deviation of elevation within a circular focal area with a radius x for x in 10, 20, 30, 40, and 50 pixels |
| SIE | Measure of solar radiation that incorporates slope and aspect [63] |
| SLOPE | Angle of steepest terrain decline |
| SLOPE_ASPECT_COS | Cosine of aspect multiplied by the slope |
| SLOPE_ASPECT_SIN | Sine of aspect multiplied by the slope |
| SLOPE_DERIV | First derivative of slope, representing the rate of change of slope |
| TPI_x | Difference between elevation at a given location and mean elevation of an annulus surrounding that location with an outer radius of x for x in 10, 20, 30, 40, and 50 pixels, where the inner radius is equal to x/2 [65] |
| TRAI | Measure of solar radiation that only incorporates aspect [63] |
| TWI | Wetness measure that incorporates slope, direction of water flow, accumulation of water flow, and upslope contributing drainage basin size [66] |

In addition to terrain data, we sought to understand the extent to which climate data can be used to predict BWA infestation. However, most climate data are only available at relatively coarse resolutions. To ensure consistency between all predictor data (Landsat and DEM data both have 30 m spatial resolution), ClimateNA was used to derive a suite of 30 m resolution locally downscaled annual and seasonal climate predictor variables (Table 4) [67]. Each variable was generated based on decadal means from 2011 to 2020 in order to represent the climatic time frame most closely associated with recent infestation. In total, there were 80 climate predictor variables.

### 2.5. Modeling and Accuracy Assessment

In the interest of understanding how to best map BWA infestation severity, we tested three different types of predictive models. The first was a spectral-data-only model, which is based purely on remote sensing data. Our hypothesis was that this model would be best at capturing vegetation structural and health change over time given its pure reliance on spectral change, although perhaps it may suffer from confusion with other agents of vegetation change. The second was a terrain- and climate-data-only model. Our hypothesis was that this model would be best at capturing site and climatic influences on BWA infestation severity, although perhaps at the expense of identifying local patterns of tree damage that are driven by factors not solely related to environmental conditions, such as local tree and stand structure characteristics. The third was a combined model that would ideally leverage the strengths of the first two models, and it was able to identify local variations in vegetation change while also attributing that vegetation change to BWA given regional trends in abiotic factors that support infestation.

**Table 4.** Climate variables derived from ClimateNA used in the prediction of BWA infestation severity, from https://climatena.ca/ (accessed on 1 June 2023). Seasonal abbreviations include SP = spring; SM = summer; AT = autumn; and WT = winter.

| Abbreviation | Description |
|---|---|
| AHM | Annual heat-moisture index (MAT + 10)/(MAP/1000) |
| BFFP | The day of the year on which FFP begins |
| CMD | Hargreaves climatic moisture deficit (mm), annual |
| CMD_x | Hargreaves climatic moisture deficit (mm) for each season x in (SP, SM, AT, and WT |
| CMI | Hogg's climate moisture index (mm), annual |
| CMI_x | Hogg's climate moisture index (mm) for each season x in SP, SM, AT, and WT |
| DD1040 | Degree days above 10 °C and below 40 °C |
| DD18 | Degree days above 18 °C, cooling degree days, annual |
| DD18_x | Degree days above 18 °C, cooling degree days for each season x in SP, SM, AT, and WT |
| DD_18 | Degree days below 18 °C, heating degree days, annual |
| DD_18_x | Degree days below 18 °C, heating degree days for each season x in SP, SM, AT, and WT |
| DD5 | Degree days above 5 °C, growing degree days, annual |
| DD5_x | Degree days above 5 °C, growing degree days for each season x in SP, SM, AT, and WT |
| DD_0 | Degree days below 0 °C, chilling degree days, annual |
| DD_0_x | Degree days below 0 °C, chilling degree days for each season x in SP, SM, AT, and WT |
| EFFP | The day of the year on which FFP ends |
| EMT | Extreme minimum temperature over 10 years (°C) |
| EREF | Hargreaves reference evaporation (mm), annual |
| EREF_x | Hargreaves reference evaporation (mm) for each season x in SP, SM, AT, and WT |
| EXT | Extreme maximum temperature over 10 years (°C) |
| FFP | Frost-free period |
| MAP | Mean precipitation (mm), annual |
| MAT | Mean temperature (°C), annual |
| MCMT | Mean coldest month temperature (°C) |
| MSP | Mean summer precipitation (mm) |
| MWMT | Mean warmest month temperature (°C) |
| NFFD | The number of frost-free days, annual |
| NFFD_x | The number of frost-free days for each season x in SP, SM, AT, and WT |
| PAS | Precipitation as snow (mm), annual |
| PAS_x | Precipitation as snow (mm) for each season x in SP, SM, AT, and WT |
| PPT_x | Mean precipitation for each season x in SP, SM, AT, and WT |
| RH | Mean relative humidity (%), annual |
| RH_x | Mean relative humidity (%) for each season x in SP, SM, AT, and WT |
| SHM | Summer heat-moisture index (MWMT)/(MSP/1000) |
| TAVE_x | Mean average temperature for each season x in SP, SM, AT, and WT |
| TD | Temperature difference between MWMT and MCMT, or continentality (°C) |
| TMAX_x | Maximum average temperature for each season x in SP, SM, AT, and WT |
| TMIN_x | Minimum average temperature for each season x in SP, SM, AT, and WT |

The modeling algorithm of choice was random forests given its capacity to handle large numbers of predictor variables, its valuable internal measures of variable importance, its robustness against overfitting and collinearity among predictors, its inability to make extreme predictions outside of the range of observed values, and its ability to model non-linear relationships between predictor and response variables [68]. Random forests have been demonstrated to be effective in previous forest degradation mapping studies (e.g., [69]). All modeling was carried out in R with particularly heavy reliance on the sf, terra, randomForest, ranger, and VSURF libraries [70–74]. The three types of models (spectral-only, terrain/climate-only, and combined) were built in precisely the same manner. They first underwent a variable selection procedure to reduce a large number of predictors to a smaller, more meaningful set with greater predictive power and less noise. To carry this out, we used variable selection using random forests (VSURF) algorithm created by Genuer et al. [71] with all 58 observations. We then tuned important random forest model parameters, including the number of variables to test at each tree split (mtry) and the minimum node size using the tuneRanger algorithm by Probst et al. [75], again using all

observations. The resulting hyperparameters can be observed in Figure A1. We did not tune the number of trees and selected 500 for each model to balance model robustness with the computational expense. For prediction and mapping purposes, we used all observations to build three optimized random forest models and applied the models to make predictions, resulting in three severity maps: one driven by spectral data alone; one driven by terrain and climate data alone; one driven by spectral, terrain, and climate data combined.

Even though random forests tend to avoid overfitting, it is still important to evaluate model performance using test data that are not used in model construction. To that end, in a separate analysis, we reconstructed the model using a leave-one-out cross-validation procedure. However, spatial autocorrelation in the dependent variable can act to artificially inflate the measures of model performance if not accounted for [76]. Accordingly, we used a buffered leave-one-out cross-validation framework, whereby on each iteration, a test plot was buffered by a given distance and all plots outside of that buffer were used to train a model. To understand how robust these models were relative to spatial effects, we buffered them by four distances: 0 km (i.e., the same as regular leave-one-out cross-validation with no consideration for spatial autocorrelation), 2 km, 4 km, and 6 km. We assessed the model's performance using the coefficient of determination ($R^2$) and root mean squared error (RMSE).

To understand which spectral, terrain, and climate variables were most important for predicting BWA infestation severity, variable importance was evaluated as the proportional error that would result if each of the selected variables were removed from consideration. To understand landscape-scale drivers and geospatial indicators of BWA infestation, we computed the accumulated local effects of each of the most important predictors within the three models and interpreted the resulting trends [77].

### 2.6. Generating a Mapping Mask

To maximize the accuracy and utility of the resulting infestation severity map data, it was necessary to generate a mask within which the results were most relevant. First and foremost, this required the development of a mask that represents areas where the primary BWA host species, subalpine fir, was present. We explored several existing datasets that could serve that function, but for a variety of reasons, we opted to generate our own mask, and the reasons include the following: (1) existing vegetation-type maps lacking species-level data, instead representing dominant species assemblages (e.g., USDA Forest Service Vegetation Classification, Mapping, and Quantitative Inventory mid-level vegetation-type maps and LANDFIRE Existing Vegetation Type); (2) lack of temporal relevancy (e.g., 2002 USDA Forest Service Individual Tree Species Parameter Maps); and (3) the tendency for local-scale vegetation map products to be more accurate in a particular area than national-scale maps, given that they are trained solely on local data and can be tailored to local geographic conditions. Although this was an important step in the analytical process, for the sake of brevity, the methods and results for developing this tree species presence/absence mask have been reported in Appendix A.

The resulting map required further refinement in order to account for other non-BWA-related disturbances, as these disturbances would likely appear as false positives in the eventual BWA infestation severity map. There were three primary disturbances of interest: (1) fire, (2) forest management activities, and (3) wind throw. To mask out fires, we used 2013–2022 fire perimeter data from the National Interagency Fire Center. All areas within fire perimeters were removed from the final map. To mask out forest management activities, we used the USDA Forest Service Activity Tracking System (FACTS) database. We only masked out activities that were attributed as having been completed between 2013 and 2022 and only focused on activities that had some component of vegetation clearing. Note that only activities on federally owned lands are reflected in the FACTS database. Forest management activities on other land ownership types within UWCNF boundaries are not included and therefore are not masked. With respect to wind throw, while this may ordinarily be a minor disturbance to consider, there was a major wind event in the

region on 8 September 2020, which resulted in large swaths of blowdowns, particularly in subalpine fir forests. A wind throw mask was manually digitized using the interpretation of high-resolution aerial image data from the 2021 National Agricultural Inventory Program with the assistance of a Sentinel-2 2020–2021 NDVI image difference for identifying local hotspots of vegetation change.

*2.7. Comparison to Aerial Survey Data*

To gain unbiased insight into the performance of our modeling and mapping approach, we compared our results to an independent aerial survey dataset. For decades, the USDA Forest Service Aerial Detection Survey (ADS) program has conducted systematic aerial surveys aimed at identifying, spatially delineating, quantifying the severity of, and attributing causal agents to forest degradation [78]. The data, collected during flight at considerable altitude and speed, are not without their limitations, including spatial uncertainty resulting from manual polygon delineation and thematic uncertainty resulting from surveyor bias and the difficulty in distinguishing between host species and causal agents [79,80]. Furthermore, some years lack complete coverage, such as in 2020 when COVID-19 limited the capacity of aerial surveys [81]. Accordingly, comparisons to other field- or remote-sensing-based data must be carried out cautiously with those limitations in mind.

To compare our results to ADS data, we acquired all ADS damage polygons between 2017 (when BWA was first identified in Utah) and 2022 (the most recent year of available data) within the UWCNF. We filtered the data to only focus on polygons attributed to subalpine fir as the host species and BWA as the damaging agent. The most directly comparable attribute within the dataset to our severity rating system was the percent affected, which describes the proportion of a damaged polygon that experienced some form of degradation in one of five classes: (1) very light (1%–3%); (2) light (4%–10%); (3) moderate (11%–29%); (4) severe (30%–50%); and (5) very severe (>50%). Given that the same areas were frequently resurveyed, it is common for there to be multiple overlapping polygons with different ratings. To address this, in overlapping areas, we used the most severe percent affected class. We compared each of the resulting polygons to the mean within-polygon severity from our combined model's map to qualitatively compare the ADS percent affected class to our map's results. We also carried out qualitative comparisons between the spatial distribution of ADS percent affected classes and the distinction between BWA and other subalpine fir damage agents and our maps.

## 3. Results

### 3.1. Field Data

In total, 58 field plots were collected between 2021 and 2022 (Figure 4), containing a total of 4441 trees, 54% of which were subalpine fir. Of the 2409 live subalpine fir trees, 1502 (62%) had at least some indication of BWA infestation (*BDS* > 0). Of the 254 newly dead subalpine firs (needles still present; post-mortality), 132 (52%) had evidence of BWA. The smaller proportion of infested new dead trees than live infested trees may be due to the relatively recent onset of infestation in this study area and the time it takes for BWA to cause mortality. Other damage agents (including bark beetles, fir broom rust, pathogens, western spruce budworm, and other mechanical damage) were present in 1166 (48%) live and 244 (96%) new dead subalpine fir trees. Proportionally, other agents played a larger role in tree mortality than BWA in our study area. However, it was quite common for multiple agents to affect trees, with 584 (24%) live and 123 (48%) new dead trees featuring both BWA and at least one other agent. Among other agents, bark beetles played a particularly important role in tree mortality, as 226 (89%) new dead subalpine fir trees possessed evidence of bark beetle damage. We did not attempt to distinguish between different species of bark beetle, but we understand that the western balsam bark beetle (*Dryocoetes confusus* Swaine) is among the primary species that attack subalpine fir trees in the region.

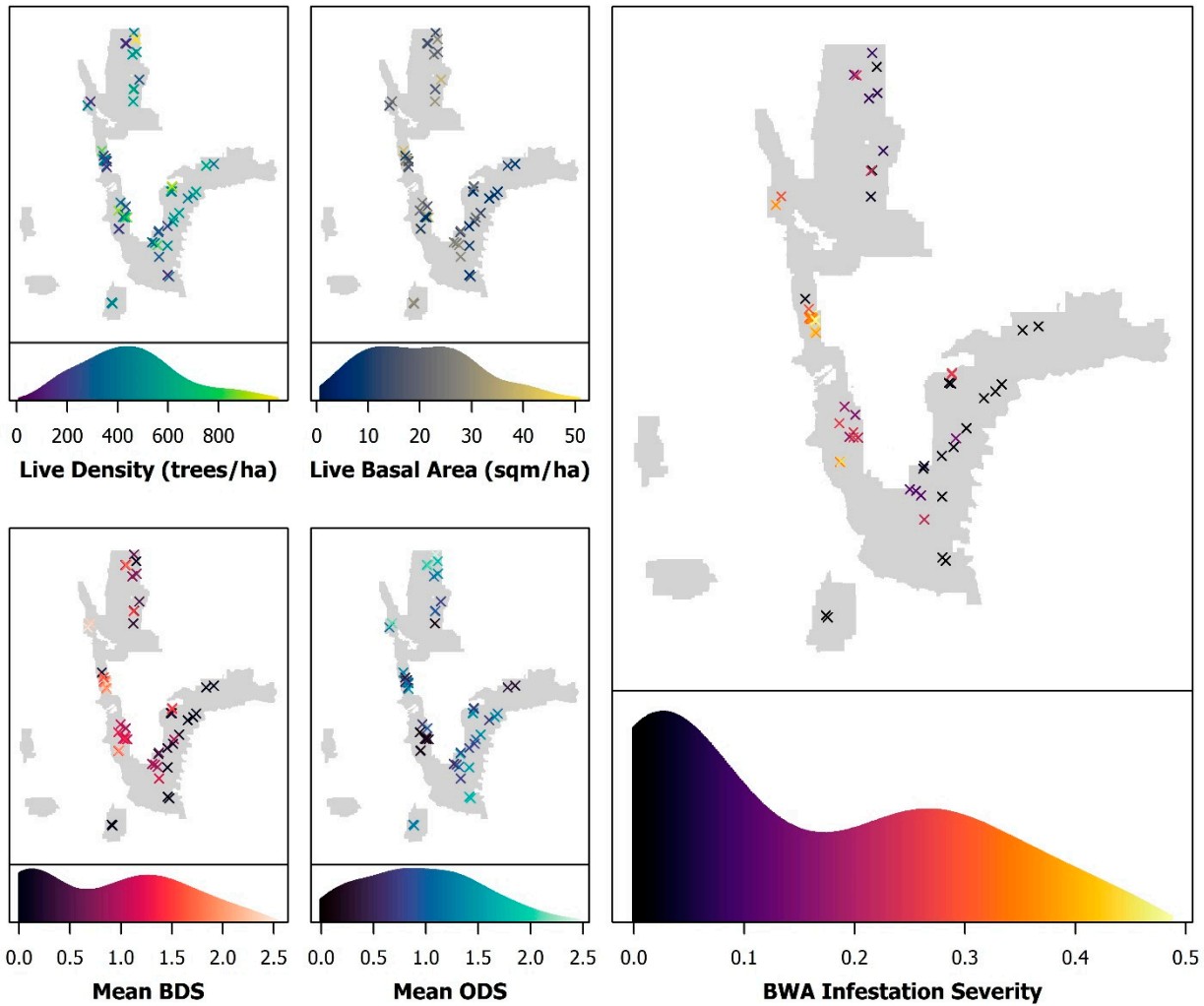

**Figure 4.** Summary of selected variables among subalpine fir trees within each field plot. For each variable, the plot points are represented as x colored by the magnitude of the variable overlaid on a map of the study area in gray. Beneath each map is a kernel density plot representing the proportional abundance of variable magnitudes among all the field plots.

The plots featured a fairly even mixture of damage agents, with 30 plots featuring higher average *BDS* (BWA was the dominant damage agent) and 28 featuring higher *ODS* (other agents were dominant). We captured a wide range of subalpine fir tree density and basal area throughout the study area (Figure 4). The spatial patterns of *BDS* and *ODS* were somewhat opposite of one another, with *BDS* generally being higher on the western portion of the study area (the Wasatch Mountains) and *ODS* being generally higher in the eastern portion (the Uinta Mountains). Field-measured BWA infestation severity was the highest in the central Wasatch mountains, low–moderate in the northern Wasatch and western Uintas, and absent in the higher elevation central and northern Uintas.

Figure 5 depicts a selection of our 15 m radius field plots drawn over 2021 high-resolution aerial imagery from the National Agricultural Inventory Program. Figure 5A–C represent some of the most severe plots in our database, Figure 5D–F represent low–moderate infestation severities, and Figure 5G–I represent non-infested plots. These images highlight the subtlety of BWA infestation symptoms within a stand and the challenges associated with characterizing severity with remote sensing data. Figure 5B, for example, appears to be a largely healthy forest despite its high severity rating. Conversely, Figure 5G clearly contains a number of standing dead trees yet featured no BWA presence at all.

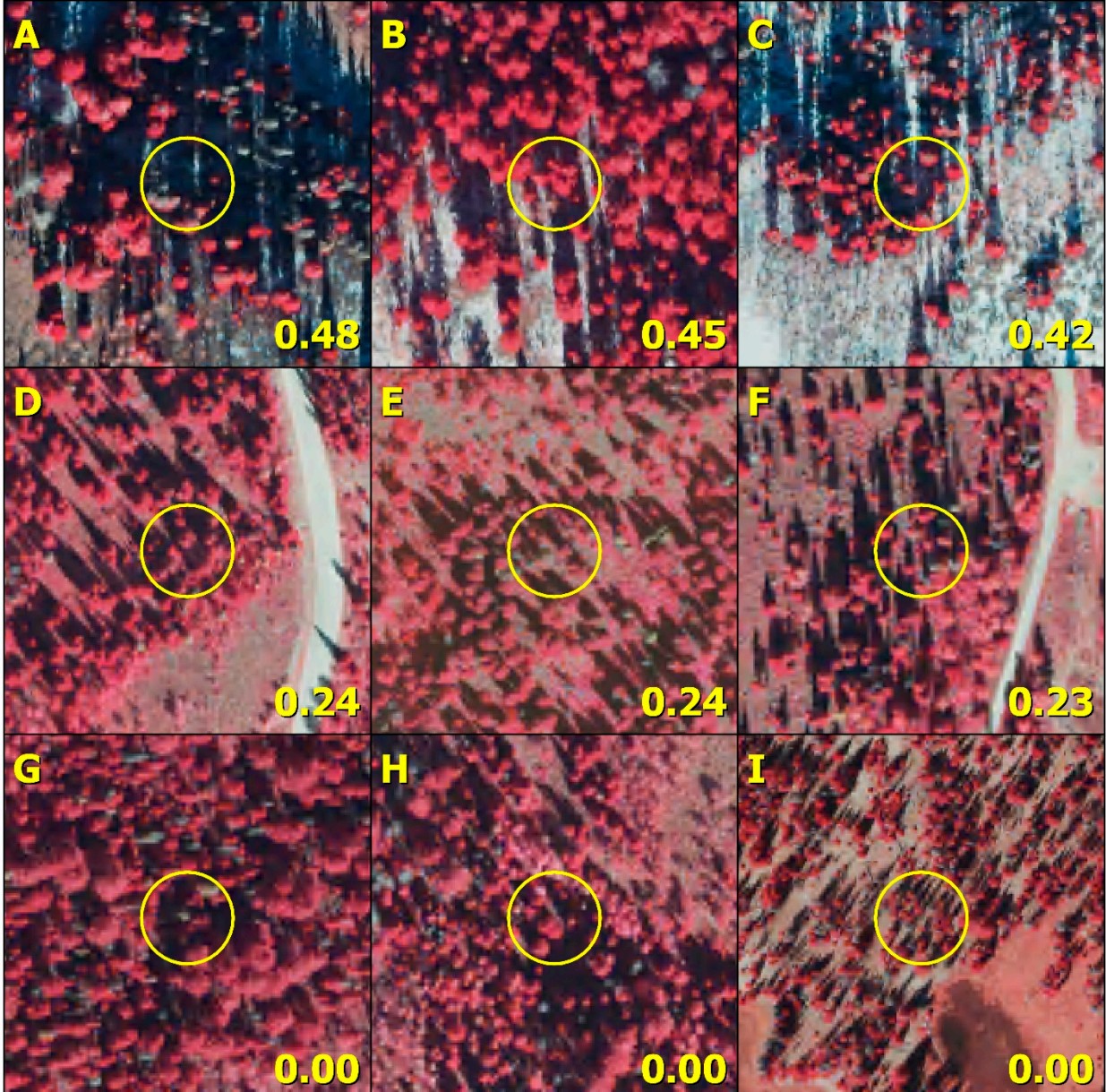

**Figure 5.** Nine examples of our 15 m radius field plots overlaid on 2021 National Agricultural Inventory Program (NAIP) image data, including three high-severity plots (**A–C**), three low–moderate severity plots (**D–F**), and three non-infested plots (**G–I**). Imagery is displayed as a color-infrared composite (R = near-infrared reflectance; G = red reflectance; B = green reflectance) to highlight differences in tree health. Numbers in the bottom right of each map represent the infestation severity for the plot shown.

### 3.2. Model Results

All three BWA severity predictive models performed well, with most explaining over 50% of the variance in plot-level infestation, but important differences emerged between the three modeling approaches (Figure 6). The spectral-only model performed the worst out of the three. At best, it explained just over half of the variance in severity ($R^2_{0km}$ = 0.541) and was able to make predictions with an RMSE of 0.105, but this is based on the 0 km buffer leave-one-out cross-validation, which fails to account for spatial effects. Indeed, the spectral model clearly demonstrates influence from fine-scale spatial autocorrelation, as increasing buffer sizes reduce model performance in terms of both explained variances ($R^2_{2km}$ = 0.507; $R^2_{4km}$ = 0.489; $R^2_{6km}$ = 0.391) and predictive error (RMSE$_{2km}$ = 0.109; RMSE$_{4km}$ = 0.111;

$\text{RMSE}_{6km} = 0.121$). The terrain- and climate-only model performed significantly better than the spectral-only model at all buffer distances, with an average $R^2$ of 0.746 and RMSE of 0.078 across all buffer distances. The combined model performed the best of the three at all buffer distances, with an average $R^2$ of 0.836 (12% better than terrain and climate and 47% better than spectral) and RMSE of 0.065 (17% better than terrain and climate and 42% better than spectral). Neither the terrain and climate nor combined models suffered greatly from spatial effects, as even at the largest buffer distance (6 km), the model's performance was not substantially worse than cross-validation without a buffer. All models at all buffer distances tended to overpredict low severities and underpredict high severities, as indicated by slopes of less than one on the regression lines representing the relationship between modeled (y) and measured (x) severities. We tested a quantile-based bias correction approach [82,83] to determine if this phenomenon could be minimized, but the results were only marginally beneficial to the spectral model at short buffer distances and were detrimental in all other cases, so we opted not to perform bias correction of any kind (Figure A2). The terrain and climate model had slopes closest to one, suggesting a greater ability to predict extreme values, which may be desirable in certain circumstances; however, with the greatest proportion of variance explained and the lowest predictive error, the combined model still produced the best results.

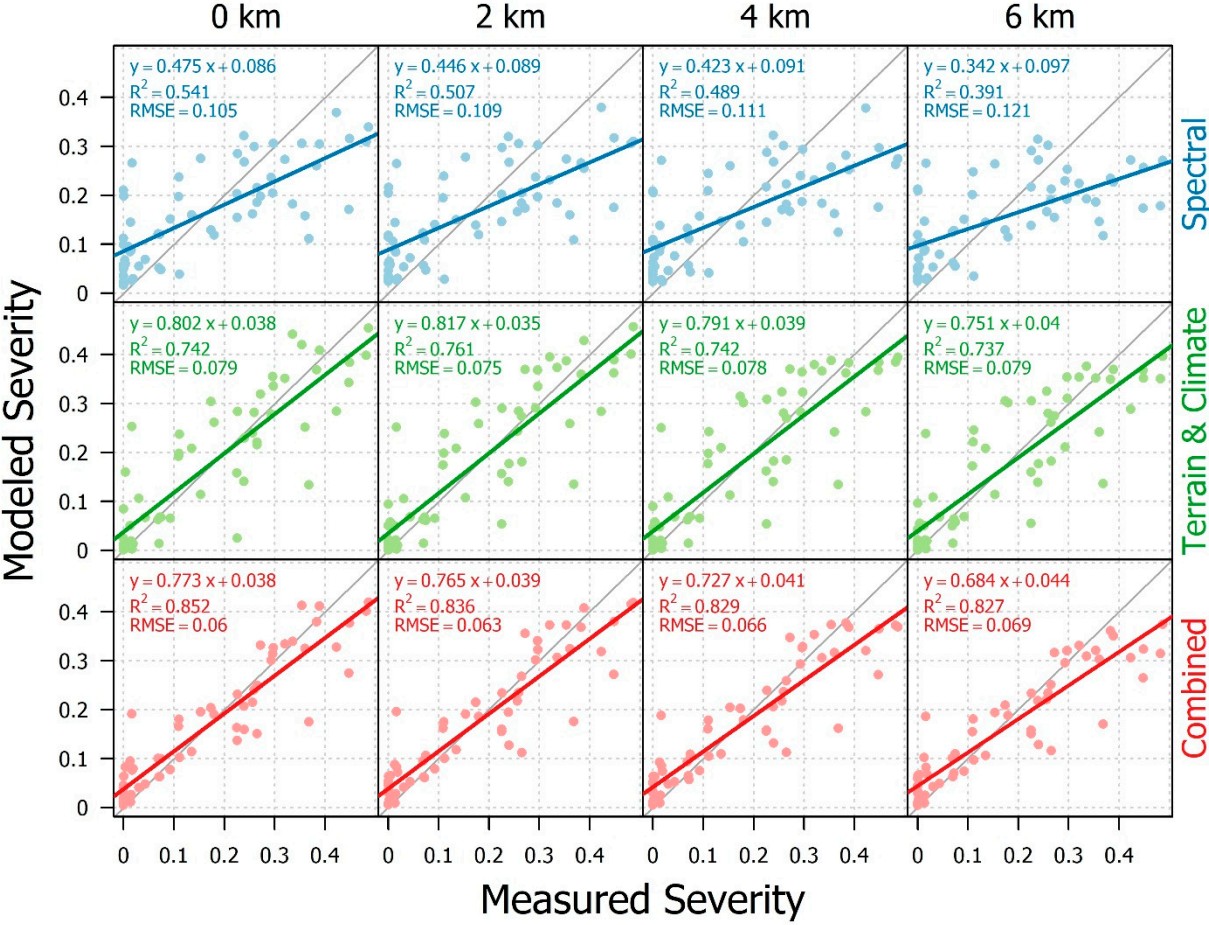

**Figure 6.** Performance of the three BWA infestation severity models (spectral, terrain, and climate, and combined) at different buffered leave-one-out cross-validation buffer distances. Points represent the modeled and measured severities resulting from the cross-validation procedure. The thin gray line represents the 1:1 relationship, showing perfect agreement between modeled and measured results. The thick colored lines represent an ordinary least squares regression between modeled (y) and measured (x) severities among the cross-validated plot data.

### 3.3. Evaluating Drivers and Predictors of Infestation Severity

The variable selection and importance results, which provide insight into the environmental drivers and geospatial predictors of infestation severity, can be observed in Figure 7. The final spectral model was reduced to eight predictors from an original set of thirty-six. Half of those predictors characterized linear trends in spectral change from 2013 to 2022 (SPEC_SLOPE_B2, SPEC_SLOPE_B1, SPEC_SLOPE_B3, and SPEC_SLOPE_TCW). Figure A3 shows the time series for the four highest and four lowest severity plots' time series data and associated linear trends for these four predictors. According to the accumulated local effects plots (Figure 8), severity was linked to positive trends in the visible wavelengths (B1 = coastal aerosol, 0.43–0.45 μm; B2 = blue, 0.45–0.51 μm, B3 = green, 0.53–0.59 μm), which we attribute to decreases in the absorption of photosynthetically active radiation in heavily infested stands over time. Conversely, decreases in tasseled cap wetness (TCW), which may represent losses in canopy moisture, were more prevalent in high-severity plots. The other half of the predictors represent 2013 spectral conditions, all of which point to the fact that higher vegetation index values (NDVI, MSAVI, SAVI, and TCW) were associated with higher infestation. We suspect that this captures regional variations in vegetation structure and productivity, with higher-elevation, cooler-temperature forests of the less-infested Uinta Mountains having a shorter growing season and more standing dead vegetation from previous spruce and pine beetle kill events, as compared to lower-elevation warmer-temperature forests of the more highly infested Wasatch Mountains.

The final terrain and climate model was reduced to two predictors from an original set of eighty. The two variables selected were CLIM_TMIN_SM (minimum average summer temperature) and CLIM_DD_0_AT (degree days below 0 °C or "chilling degree days" in Autumn) (Figures 7 and 9). Both are temperature-related, and neither are precipitation related. Notably, not a single terrain metric was selected. However, ClimateNA's climate downscaling procedure is driven by elevation data. Thus, the terrain is implicitly important, although it is not as valuable as the terrain-based climate products. The accumulated local effects plots revealed the following relationships: Currently, severity is highest in areas with (1) minimum summer temperatures greater than 8 °C and (2) autumn chilling degree days less than 80. Both indicate temperature's influence on BWA population establishment and growth toward warmer areas and/or areas that have shorter cold periods. Scatterplots comparing these two climate variables to infestation severity at the plot level can be observed in Figure A4.

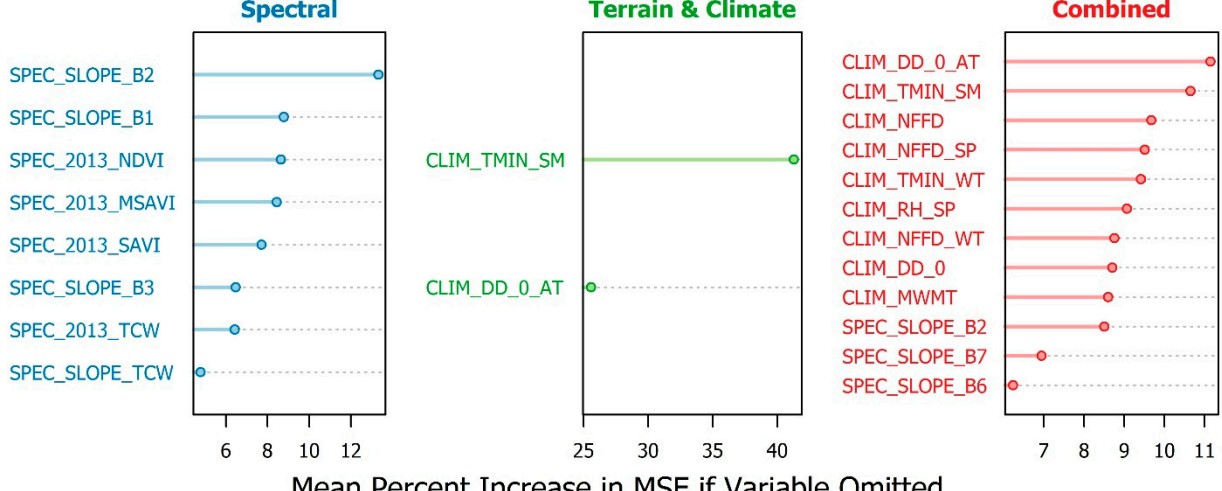

**Figure 7.** Variable importance of the three BWA infestation severity models (spectral, terrain and climate, and combined).

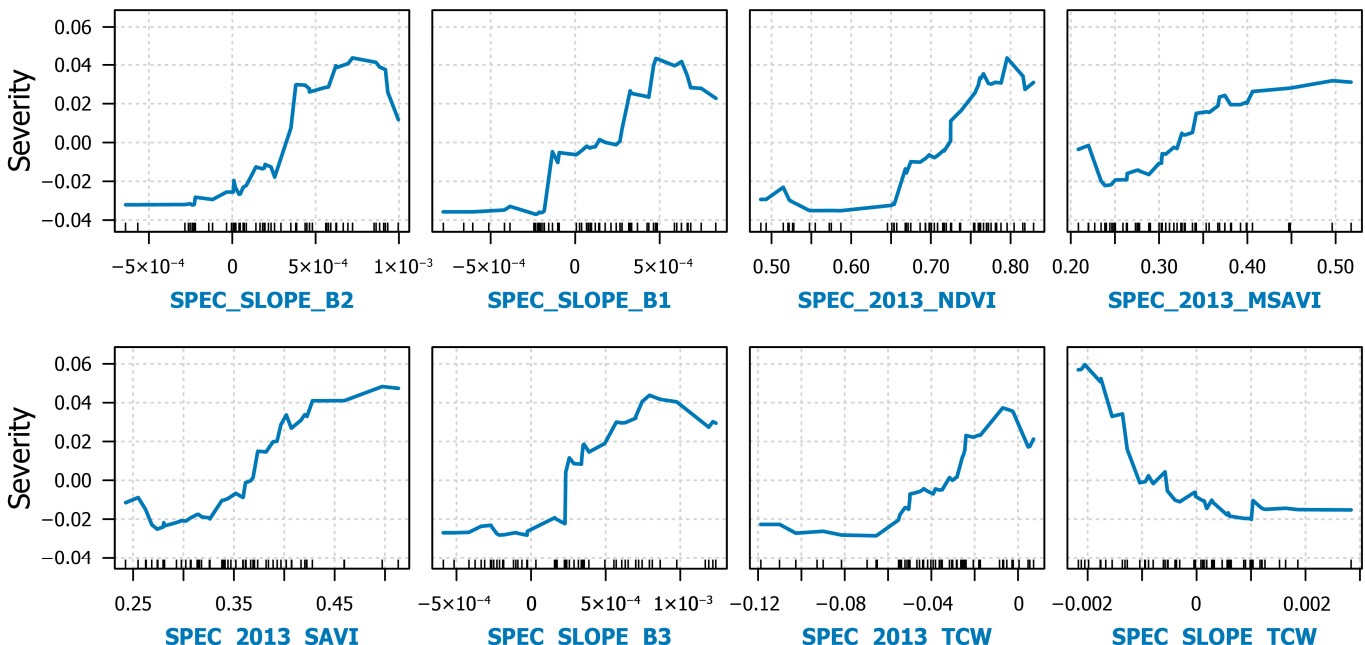

**Figure 8.** Accumulated local effects plots conveying the relationships between the most important spectral predictors and BWA infestation severity in the spectral-only model. Vertical black lines atop the *x*-axis represent observed predictor values among the field plots. Abbreviations: B1 = Landsat band 1; B2 = Landsat band 2; B3 = Landsat band 3; NDVI = normalized difference vegetation index; MSAVI = modified soil adjusted vegetation index; SAVI = soil adjusted vegetation index; TCW = tasseled cap wetness; SLOPE = linear trend in spectral change over time; 2013 = spectral values at the start of the time series.

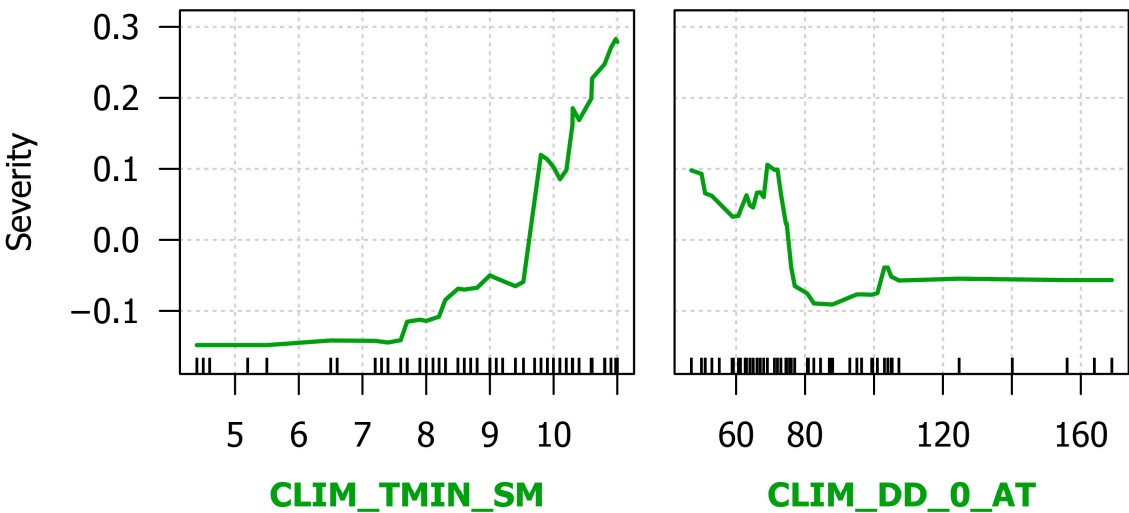

**Figure 9.** Accumulated local effects plots conveying the relationships between the most important climate and terrain predictors and BWA infestation severity in the terrain- and climate-only model. Vertical black lines atop the *x*-axis represent observed predictor values among the field plots. Abbreviations: TMIN = minimum temperature; DD_O = degree days below zero; SM = summer; AT = autumn.

The combined model was reduced to 12 predictors from an original set of 116 (Figures 7 and 10). Climatic variables comprised the top nine, and spectral trends over time comprised the bottom three in terms of variable importance. Climatically, the same two variables as the terrain and climate model were the most important, with nearly identi-



cal accumulated local effects relationships relative to severity. Additionally, the number of frost-free days annually (CLIM_NFFD), in spring (CLIM_NFFD_SP), and in winter (CLIM_NFFD_WT) were also selected, all of which featured positive relationships with severity such that increased frost-free days promoted higher infestation severity. Higher minimum winter temperatures (CLIM_TMIN_WT) and mean warmest month temperatures (CLIM_MWMT) were both related to higher severity. Conversely, lower annual degree days below zero (CLIM_DD_0) were associated with higher infestation severity. All aforementioned variables are temperature-related, and all point toward the same trend: Generally, warmer areas promote higher infestation. The only climatic variable that related to precipitation was relative humidity in spring (CLIM_RH_SP), which was positively related to infestation. The three spectral trends (SPEC_SLOPE_B2, SPEC_SLOPE_B7, and SPEC_SLOPE_B6) were all positively related to infestation such that increased reflectance in these spectral regions (B2 = blue, 0.45–0.51 μm; B6 = shortwave infrared 1, 1.57–1.65 μm; B7 = shortwave infrared 2, 2.11–2.29 μm) was linked to high severity. We attribute these relationships to decreases in photosynthetic absorption (B2) and canopy water (B6 and B7) absorption resulting from infestation.

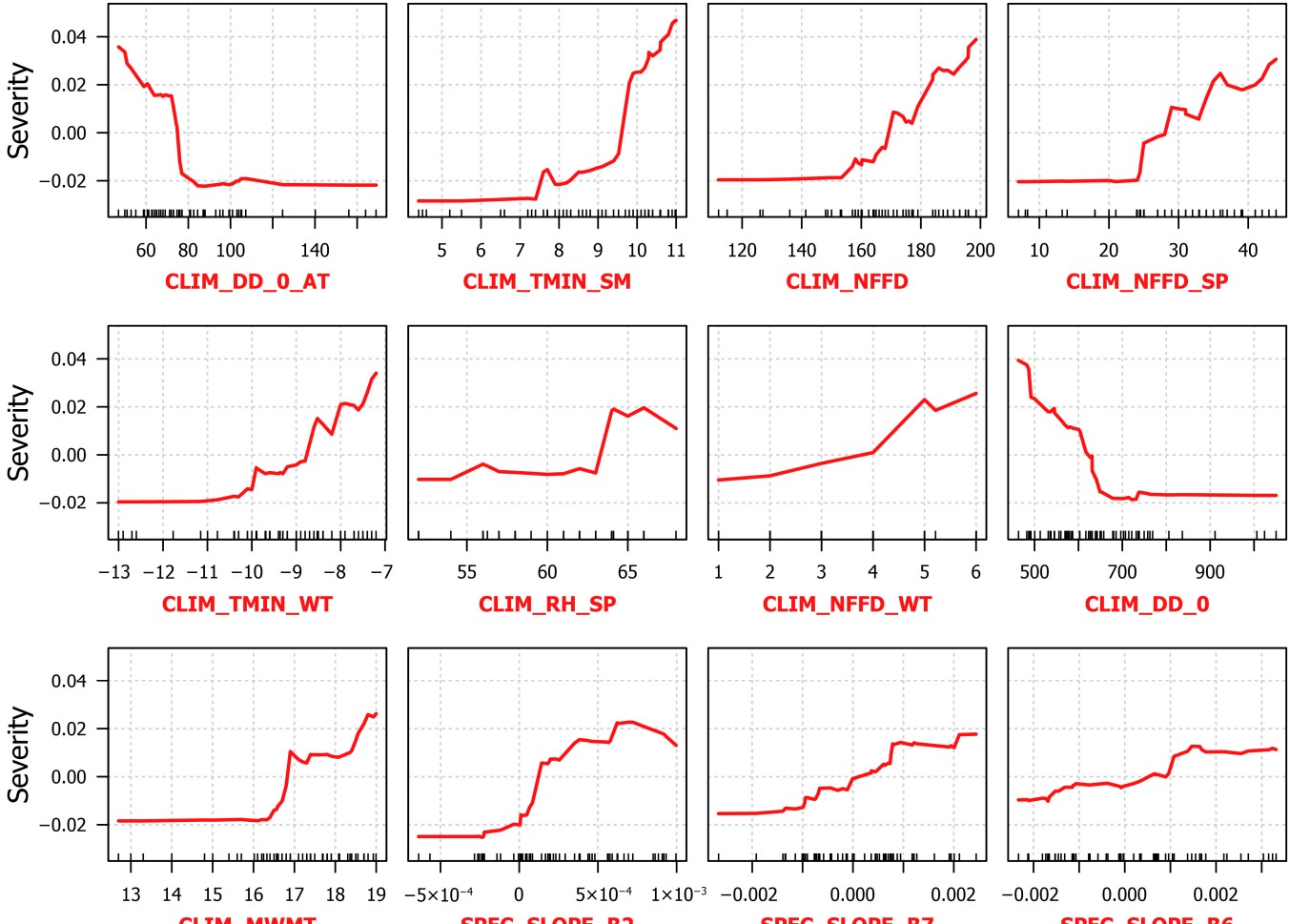

**Figure 10.** Accumulated local effect plots conveying the relationships between the most important spectral, climate, and terrain predictors and BWA infestation severity in the combined model. Vertical black lines atop the *x*-axis represent observed predictor values among the field plots. Abbreviations: CLIM = climate variables; SPEC = spectral variables; AT = autumn; SP = spring; SM = summer; WT = winter; DD_0 = degree days below zero; TMIN = minimum temperature; NFFD = number of frost-free days; RH = relative humidity; MWMT = mean warmest month temperature; SLOPE = linear trend of spectral variables; B2 = Landsat band 2; B6 = Landsat band 6; B7 = Landsat band 7.

### 3.4. Map Results

Mapping these three models across all subalpine fir forests in the UWCNF reveals broader spatial trends that cannot be accounted for in the aspatial scatterplots of a relatively small sample of predictions vs. observations. Figure 11 compares the mapped results of all three models, where important differences emerge. The spectral map depicts an overall more moderate distribution of infestation severity, with fewer areas containing very high or very low infestation, but most of the UWCNF possesses some degree of infestation. We attribute this to the spectral model's relative inability to capture variations in measured severity and confusion with other damage agents, resulting in a model that tends to make predictions closer to the mean. The terrain- and climate-only map features a much starker spatial contrast of high versus low severity areas with a relatively higher infestation in the Wasatch Mountains (north–south-oriented mountain range on the west side of the contiguous forest boundary) and lower infestation in the Uinta Mountains (east–west-oriented mountain range to the east). The Uinta Mountains reach higher elevations and have generally cooler temperatures even at comparable elevations to the Wasatch Mountains. Given the apparent temperature dependency of the BWA described earlier, terrain and climate maps clearly reflect regional differences in climate-driven BWA infestation patterns. As expected, the combined model features characteristics of both the spectral map and the terrain and climate map. For example, the spectral map captures local-scale variability in forest degradation, whereas the terrain and climate map captures broader-scale variability in environmental conditions that promote BWA infestation. Thus, the combined map contains both patterns, characterizing local- and regional-scale BWA-driven degradation very effectively within our study area. However, given the influence of the spectral trends on the combined model suggesting that there is some degree of forest degradation occurring in the Uinta Mountains, the combined model does result in some false-positive characterization of low-level infestation in that range. The infestation map values in that region (generally <0.05) are lower than the average prediction error of the combined model (mean RMSE = 0.065), suggesting that they are within the typical margin of error and should not be treated as reliable.

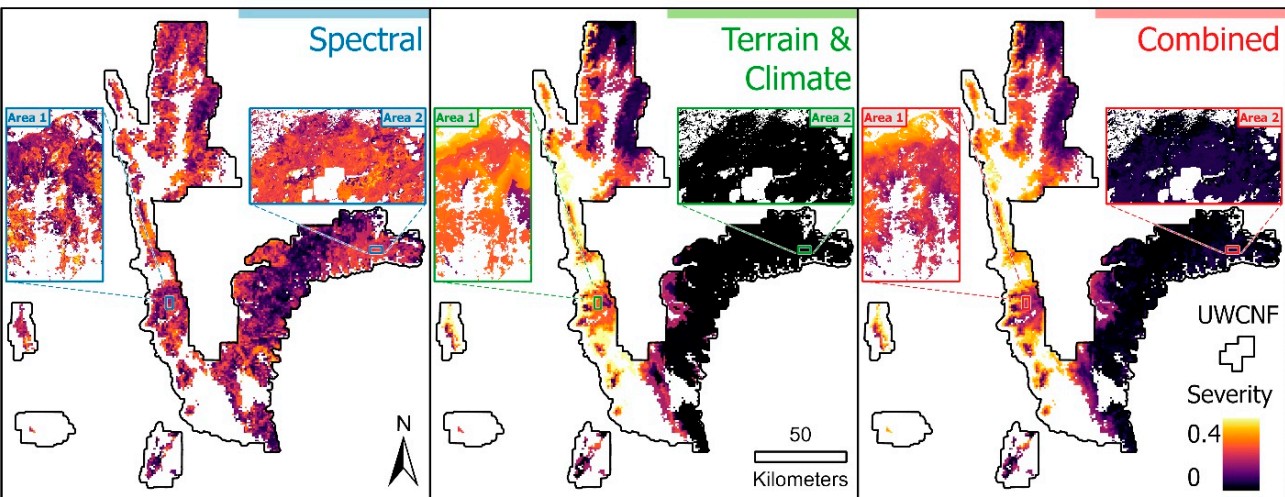

**Figure 11.** Comparison between BWA infestation severity prediction maps based on three different models (spectral, terrain and climate, and combined). Pixels have been aggregated to a 900 m resolution median from their original 30 m resolution to enhance visual interpretation on the main maps. The focus maps (Area 1 and Area 2) are displayed using the original 30 m resolution.

### 3.5. Comparison to Aerial Survey Data

A comparison between our maps and aerial survey data revealed general agreement both quantitatively and spatially (Figure 12). Comparing the five ADS BWA percent affected classes to our combined map, for example, yielded a clear positive relationship

(Figure 12A), where higher-severity ADS classes tended to be mapped as having higher severity using our combined model. There was, however, a fair amount of spread among the mapped severities within each ADS class. The spatial trend in ADS BWA severity classes qualitatively matched those of our maps, with a higher infestation in the central Wasatch Mountains that decreased radially outwards, eventually reaching complete BWA absence in the central and eastern Uintas (Figure 12B). Figure 12C highlights how BWA is only one of several agents acting to degrade subalpine fir forests. Of the 4952 ADS polygons labeled as subalpine fir between 2017 and 2022, 60% were attributed to BWA as the damaging agent, 20% were attributed to subalpine fir decline, 18% were attributed to root disease and beetle complex, 2% were attributed to western spruce budworm, and <1% were attributed to twig beetles. The spatial distribution of the other agents, in particular, helps explain in part why the spectral model was identifying forest degradation in the Uintas.

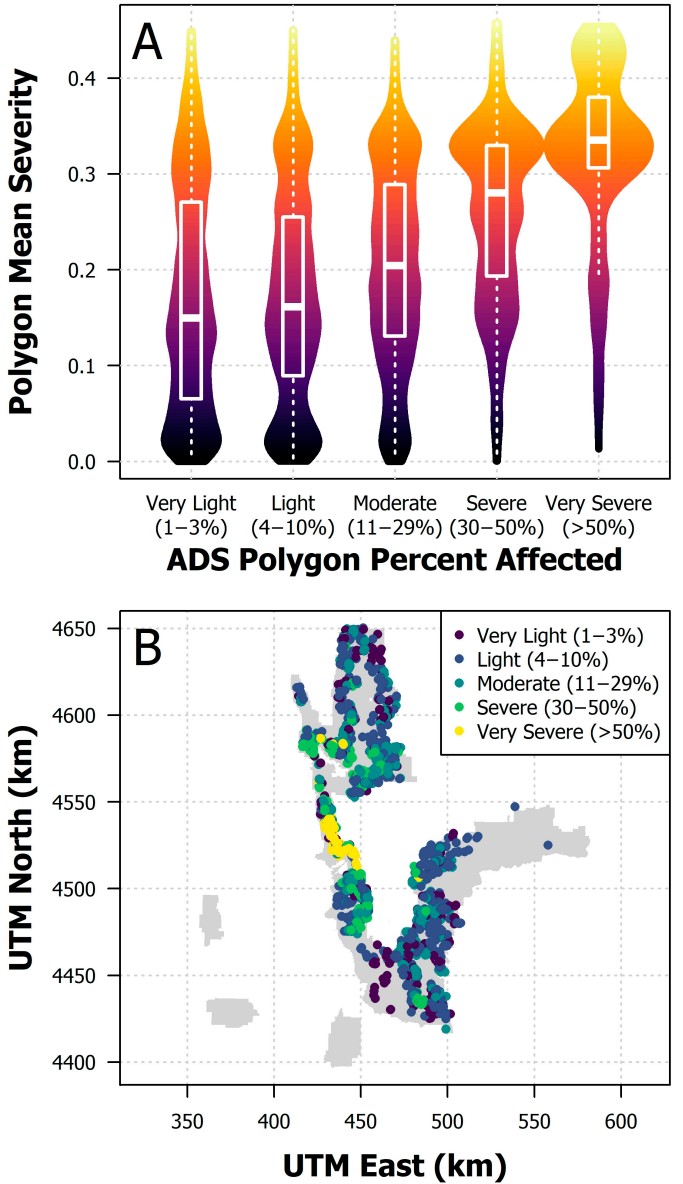

**Figure 12.** *Cont.*

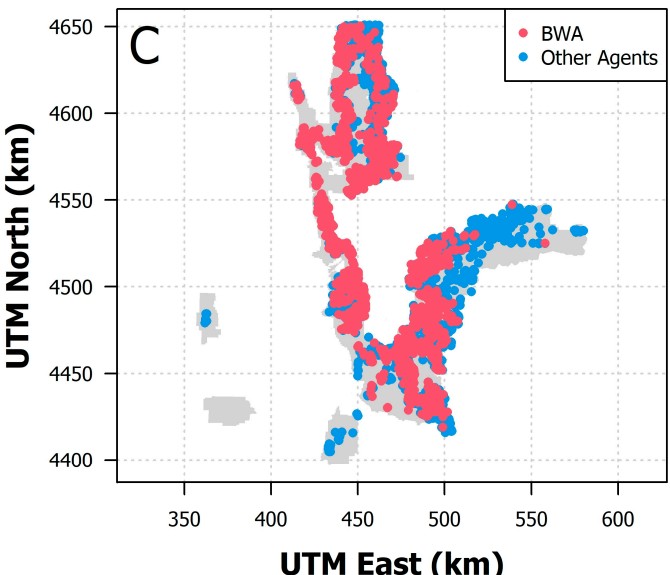

**Figure 12.** Results of the comparison between our combined model's map of severity and ADS data compiled within the study area from 2017 to 2022, including the following: (**A**) comparison between ADS polygon percent affected classes versus within-polygon mean severities from our map; (**B**) map of BWA-specific percent affected classes within our study area; and (**C**) map comparing the spatial distribution of BWA and other agents within our study area.

## 4. Discussion

In this study, we aimed to map the severity of an ongoing and relatively recent invasion of BWA into the subalpine fir forests of UWCNF in northern Utah. Given the limited research studies on mapping BWA infestation, we sought to test three different modeling approaches for deriving the most accurate, useful map for land managers. The spectral-only model, based solely on a time series of Landsat imagery, performed the worst of the three, although it was still able to account for approximately half of the variance in severity. We attribute the comparably poor performance to the fact that image data alone have a difficult time discerning between agents of change. Spectral reflectance and vegetation indices are excellent at identifying the extent and timing of disturbances and can certainly distinguish some types of disturbance from others (e.g., fire vs. harvesting) based on the differences in pre- and post-disturbance spectral characteristics [44,84,85]. However, identifying the subtlety of BWA-induced damage, which occurs over several-year timescales, and distinguishing it from more pronounced but spectrally similar damage, such as the bark-beetle-induced mortality of subalpine fir or other codominant tree species, proved to be a challenging endeavor. Figure 5 illustrates this challenge, providing examples of heavily infested forests that appear healthy and vice versa among our field plots. Nearly half of our plots exhibited damage that was dominantly attributable to non-BWA agents, further highlighting the complexity of teasing out causal factors in tree health decline in these forests. That said, our proportional agent scoring system, in concert with our approach for masking out disturbances such as fire, harvesting, and windthrow, enabled the mapping of BWA-specific infestation severity with some success, even using spectral data alone as the basis of predictions.

It is possible that the spatial and spectral resolutions of Landsat imagery may have been limiting factors to the performance of the spectral-only model. Within each 30 m pixel, even highly infested stands featured a range of canopy structures, mixtures of healthy and unhealthy trees, the presence of both host and non-host tree species, a diversity of understory vegetation, and variable ground surface materials, all of which can act to diminish the subtle spectral signal of infestation. The use of high-spatial-resolution imagery could reduce the amount of within-pixel mixing, potentially enabling the identification of

tree-level infestation. Although successful evidence of this is limited in studies specific to BWA [34], high-spatial-resolution image data from spaceborne and airborne platforms have proven effective when mapping the effects of other forest insects (e.g., [86–89]). Similarly, image data with higher spectral resolution than Landsat's seven bands may have offered an opportunity to evaluate more spectrally distinct absorptive or reflective characteristics of BWA infestation. Once again, BWA-specific hyperspectral studies are limited [32,34], but studies of other insects have demonstrated promise, including the widespread and related hemlock woolly adelgid [90].

The spectral-only model was also the most highly influenced by fine-scale spatial autocorrelation, as the model's performance decreased notably with increasing cross-validation buffers. This is likely due to local patterns in vegetation composition over space since vegetation cover has a dominant effect on spectral reflectance in forest environments. As a result, a model built relative to one subalpine fir-dominated stand may not be directly applicable to another stand with a somewhat different vegetation assemblage. There is some debate in the spatial statistical literature as to whether or not this buffered leave-one-out cross-validation approach provides robust performance estimates. Wadoux et al. [91] suggest that simple cross-validation (e.g., our 0 km buffer) tends to overestimate apparent model performance and buffered cross-validation tends to underestimate it. By presenting both cases, we feel that we have characterized our model's performance fairly. One fundamental assumption in all spatial statistics is that close-proximity objects are more related to one another than distant objects, so by ensuring that training and test points have some distance between them, as in our buffered approach, we attempted to avoid the artificial inflation of apparent model performance [76]. Despite the limitations of the spectral-only model, the variable importance and relationships observed among the predictor variables fall in line with expected trends in spectral response to BWA infestation and are corroborated by a body of previous literature [32,34,37].

The terrain- and climate-only model had impressive predictive power, explaining approximately 75% of the variance in BWA infestation severity with an average prediction error of 0.078 based on only two temperature-based climate predictor variables. This suggests that BWA populations and subsequent damage to trees are heavily influenced by temperature. Areas with relatively high minimum summer temperatures (the most important climatic predictor of infestation severity) suggest a positive yet limited relationship between summer temperatures and BWA success. In fact, Greenbank [10] found that the mean fecundity of BWA was the greatest from 8 to 24 °C in laboratory rearing studies, exhibiting increasing egg mortality at 26 °C and 100% egg mortality at 32 °C. Conversely, areas with colder temperature extremes and/or longer cold periods may limit both the inter- and intra-annual life cycle of BWA, thereby limiting infestation severity. This is illustrated by the importance of chilling degree days (the second most important predictor), which incorporates both the magnitude of difference in daily temperature from a baseline of 0 °C and the number of days below that same baseline. Thus, it is indicative of both extreme cold temperatures and/or extended cold periods. Greenbank [10] found that overwintering first instar BWA mortality began at −20 °C, and no adelgids survived at temperatures below −34 °C (unless the adelgids resided below the snowline). Additionally, Greenbank [10] concluded that colder climates may not support BWA infestations in the crown but that infestations could likely persist below the snowline and spread slowly. Note that although daily minimum temperatures may occasionally drop to lethal levels for BWA, the average minimum temperatures in Figure 10 suggest that all subalpine fir forests of the UWCNF are at risk of BWA infestation. Hrinkevich et al. [39] found very similar results to ours at a coarser resolution and broader spatial scale, with summer and autumn temperatures being the strongest predictors of BWA infestation. Although Mitchell and Buffam [11] did not explicitly test climatic variables as the predictors of BWA infestation, their finding that lower-elevation sites, which we can infer were generally warmer, tended to have greater severity, which aligns with our results. Likewise, Hicke et al. [92] found that warmer summer temperatures were associated with increased severity, mirroring our

findings. One of the great benefits of our model is the fact that it can not only be applied to the fine-scale (30 m resolution) prediction of current infestation severity, as we carried out, but it can also potentially provide insight into future BWA infestation conditions. Future studies should aim to use the results we have generated to predict the potential areas of future BWA spread caused by climate change.

The combined model appeared to overcome limitations and enhance the individual strengths of the spectral-only and terrain- and climate-only models. The spectral model was good at capturing local-scale variability in forest damage (i.e., pixel-level changes in reflectance over time), but it was bad at capturing regional-scale trends with respect to BWA infestation severity due to poor change agent distinction. The terrain and climate model precisely performed the opposite, as it was driven by data downscaled to 30 m from a much coarser spatial resolution (800 m), which, itself, is a terrain-informed imputation of a sparse network of weather stations [43]. For example, the Wasatch Mountains have certainly experienced a high degree of BWA infestation severity in recent years. The terrain and climate model accurately represents this regional phenomenon, but it depicts severity as a rather smooth gradient that is driven by trends in temperature. In reality, infestation can be highly variable at the local scale with adjacent forest stands featuring somewhat different levels of severity. This type of local variability would be missed by a purely climate-driven model and more likely to be captured by a spectrally driven model. Conversely, the Uinta Mountains have largely remained non-infested to date. The spectral-only map highlights several areas of low–moderate infestation, representing false positives that are likely caused by other damage agents presenting a similar spectral signal to BWA-induced damage, as highlighted in Figures 5 and 12C. By building a model that incorporates spectral and climate data, forest degradation in BWA-prone areas is enhanced and that in BWA-resistant areas is diminished, producing a map that best captures both the local and regional patterns of BWA infestation severity. The prevailing relationships between the selected predictors of the combined model were very similar to those in the spectral and terrain/climate models. Spectrally, areas that featured increases in visible and shortwave infrared reflectance were associated with higher severity. Climatically, areas that were broadly characterized by warmer temperatures (e.g., higher minimum temperatures, lower subzero degree days, and more frost-free days) were associated with higher severity. To be sure, our combined model has not included all possible meaningful predictors of BWA infestation. It is certainly possible that the inclusion of predictors related to wind speed/direction (to understand BWA dispersal), forest canopy cover/basal area/biomass (to understand BWA forest structural preferences), and proximity to roads/development (to understand potential human-caused BWA spread), to name a few examples, could add predictive capacities to BWA mapping efforts in future studies.

All three models relied heavily on the use of a variable selection procedure known as VSURF [71]. The goals of this process were as follows: (1) to increase model parsimony, which is generally good practice in statistical modeling; (2) to eliminate noisy or unhelpful predictors among a long list of candidates; and (3) to enhance the interpretability of model results according to prevailing trends in predictor–response relationships. While VSURF has been widely demonstrated to be effective toward these ends [93], it is worth noting that the manner by which variables are eliminated could conceivably eliminate meaningful variables. For example, the third step of the three-step VSURF algorithm eliminates highly correlated variables. Inevitably, there was some degree of correlation among the many candidate predictors for each of our three models, particularly within the climate data (Figure A5). While random forests are widely understood to be robust relative to multicollinearity [94–99], unlike parametric models such as multiple linear regression, VSURF may have removed variables that were highly correlated to, but slightly less important than, the final set of selected predictors in each model. Thus, our variable importance and selection results should be interpreted as follows: The variables that were selected were important, but the variables that were omitted were not necessarily unimportant. Moreover, with respect to our interpretation of the relationships between predictors and BWA infestation severity, our use of accumulated local

effects was specifically aimed at addressing potential multicollinearity in the final predictor set for each model [77]. Once again, while multicollinearity can negatively affect the ability to interpret the meaning of model coefficients in a parametric statistical context, assessing accumulated local effects of non-parametric random forests provides robust insight into predictor–response relationships.

In recognition of the fact that 58 plots are a relatively small sample in comparison to the large area over which we were carrying out the mapping (all subalpine fir in UWCNF) and that our plot protocol is merely one approach of many that can potentially quantify BWA infestation severity, we compared our results to an independent aerial survey dataset. The prevailing trends in severity measures and spatial distributions were well aligned between our maps and the areas identified by the USDA Forest Service ADS program as having been infested by BWA. Figure 12A illustrates that ADS polygons with higher percent affected severity classes were generally mapped as exhibiting higher severity in our combined model map, although there is clearly a fair amount of spread within each class. It is impossible to determine if this is due to uncertainty in our maps or uncertainty in ADS data, but as we acknowledged in Section 2.7, the spatial and thematic uncertainties inherent in ADS data can limit their use in serving as direct reference data for remote sensing analyses [79,80,100]. Furthermore, the primary focal metric of damage in ADS data is mortality, a signal that is easily visually detected as red, orange, or brown conifer needles, which can be readily identified even from high altitudes. Although BWA infestation can kill trees, mortality was only abundant in the most severely infested stands in our field database. The driving indicators of severe infestation were gouting, crown deformities, and the presence of wool on the tree bole, all of which are much more difficult, or impossible, to identify from the air, as exemplified in Figure 5. Thus, particularly low-level BWA infestations are likely poorly represented in ADS data.

Among the ADS damage polygons in subalpine fir forests of UWCNF, BWA was the most frequently attributed causal agent. The second-most common cause of damage was subalpine fir decline (SFD), which is also referred to as the subalpine fir mortality complex. As the name suggests, SFD is not a singular agent so much as it is a confluence of relatively poorly understood agents, including climatic factors, pathogens, and insects, all acting in concert to produce widespread subalpine fir mortality at times [101,102]. In our field data, it was nearly always the case that trees heavily impacted by BWA also featured other damage agents—especially bark beetles. Our *BDS* vs. *ODS* scoring system was designed to tease out BWA-specific damage, but the unexplained variance in each of our models (especially the spectral model) can likely be attributed to SFD. Additionally, given the apparent relationship between warmer temperatures and BWA in our terrain and climate and combined models, it is certainly possible that drought may have acted as an additional source of model confusion or even predisposed trees to BWA-induced damage. Northern Utah experienced extended drought conditions for several years preceding our field campaign, and this may have played a role in weakening trees with high moisture requirements, such as subalpine fir.

From a management perspective, it is often useful to describe severity categorically (e.g., "low", "moderate", and "high"). This type of categorization formed the basis of many of our field measurements of severity. Yet, the nature of our analytical approach yielded a measure of severity on a continuous scale. This scale ranges theoretically from zero, which would indicate not a single BWA-affected subalpine fir tree within a plot, to one, which would indicate the highest levels for all infestation metrics for every subalpine fir tree within a plot and no presence of other damage agents whatsoever. The plot-level quantitative measures of severity in our database ranged from 0 to 0.49. One might consider 0.49 to be moderate (since it falls roughly halfway between the theoretical minimum and maximum values), but in fact, this represents a heavily impacted site that one would certainly call "severe" (high mortality, severe gouting, etc.). Given that there are no hard definitions of what defines different categories of BWA infestation severity at the stand level, we have only presented our results on a quantitative scale. End users of the maps (e.g., forest

managers, ecologists, and entomologists) may choose to apply their reclassification schemes to define thresholds that equate to categories of severity that suit their own needs. Of particular interest to forest managers may be the identification of low levels of infestation, as these areas may be targeted for management priority and neighboring areas may be highly susceptible to future infestation [16].

There are a few limitations that warrant further discussion. First, there is inherent subjectivity in some of the infestation metrics that we measured. For example, distinguishing between light and moderate gout severity was sometimes a difficult judgment call. To address subjectivity, the same calls were carried out by the same person for all plots, although some variability is still certainly inherent in the data. Second, there is an innate challenge to evaluating some of those same metrics. For example, gouting is most easily observed on the ends of twigs with live foliage (Figure 2). In stands where the live crowns of taller trees were not reachable from the ground level, identifying gout was difficult, particularly if gouting was light. Third, and perhaps most importantly, the results of this analysis should be treated as primarily relevant to the UWCNF, and extrapolation outside of the study area's boundary should be done cautiously. One major reason is that we have not accounted for BWA spread over time. For example, BWA has been endemic in subalpine fir forests located north and west of the UWCNF for decades or longer. Conversely, BWA has not yet been identified in areas located south and east of UWCNF. However, a purely or mostly climate-driven model makes the assumption that BWA has had an equal opportunity to inhabit everywhere, irrespective of the actual time it takes for insect populations to spread over space. Accordingly, applying our models to north and west areas with comparable climatic conditions would likely result in the underestimation of severity, and applying our models to climatically comparable areas located south and east would likely result in the overestimation of severity. This points to a unique advantage of the inclusion of spectral data in the modeling process, as areas that have experienced vegetation change are the focus, rather than theoretical habitat suitability. In summary, we have presented a local model, both spatially and temporally, that is capable of predicting BWA infestation severity within the extent of UWCNF at present. We have not presented a global model that is capable of mapping BWA infestation over broader regions over longer timescales. Future research should aim to explore the best mapping practices for capturing broader-range variability in BWA infestation, perhaps incorporating some constraints that represent insect spread over time.

## 5. Conclusions

In this study, we have presented novel insights into the best practices for measuring, modeling, and mapping BWA infestation. BWA is unique among forest insect infestations in that the symptoms are both subtle and drawn out over time, although they are still extremely damaging and potentially fatal to host trees. Thus, mapping BWA requires a tailored analytical approach. Our remote sensing analysis demonstrated that even with a large suite of unitemporal and time series metrics computed over ten years of image data with a diverse array of vegetation indices, BWA infestation was difficult to quantify using remote sensing data alone. However, our use of downscaled climate data revealed an impressive capacity for mapping infestation severity, suggesting that BWA infestation is highly temperature-dependent. In a warming climate, this likely means that the extent of infestation will grow over time, the impacts of which could be devastating for the subalpine fir forests of Northern Utah.

Our study has resulted in the production of three maps based on random forest models using three different types of data to predict BWA infestation severity: (1) spectral data; (2) terrain and climate data; and (3) spectral, terrain, and climate data combined. The first map, although it is the least accurate, may still provide value to forest managers in identifying areas of damage from a variety of agents, BWA included, in the subalpine fir forests of UWCNF. The second map provides a strong baseline understanding of the current landscape and environmental drivers of BWA infestation severity and can be used to predict areas that are sensitive to future spread. By combining spectral and climate data,

the third map is the best representation of BWA-induced damage throughout the UWCNF, leveraging the strengths and overcoming the weaknesses of spectral-only and terrain- and climate-only approaches. We suggest that these maps should be used to inform future ground and aerial surveys aimed toward monitoring the expansion of BWA's range and ultimately to identify priority areas where proactive management may mitigate future damage to one of the most abundant tree species in Utah and beyond.

**Author Contributions:** Conceptualization, M.J.C. and J.P.W.; methodology, M.J.C. and J.P.W.; software, M.J.C.; validation, M.J.C. and J.P.W.; formal analysis, M.J.C. and E.M.B.; investigation, M.J.C. and J.P.W.; resources, M.J.C. and J.P.W.; data curation, M.J.C.; writing—original draft preparation, M.J.C. and J.P.W.; writing—review and editing, M.J.C., J.P.W., and E.M.B.; visualization, M.J.C.; supervision, M.J.C. and J.P.W.; project administration, M.J.C. and J.P.W.; funding acquisition, M.J.C. and J.P.W. All authors have read and agreed to the published version of the manuscript.

**Funding:** This research was funded by the USDA Forest Service, grant number 21-DG-11046000-609. The APC was funded by the USDA Forest Service, grant number 21-DG-11046000-609.

**Data Availability Statement:** The data presented in this study are available upon request from the corresponding author.

**Acknowledgments:** We would like to thank Ryan Davis and Daniel Ott for their assistance in field data collection.

**Conflicts of Interest:** The authors declare no conflict of interest.

## Appendix A. Creation of a Subalpine Fir Map

To create a map that represents locations within the UWCNF in which subalpine fir trees are present, we used a similar modeling framework as with mapping infestation severity (variable selection, model tuning, and random forest). However, there were a few key differences. For reference data, we used FIA plots rather than our plot data [12,103]. Given the relatively sparse distribution of subalpine fir-present FIA plots within UWCNF, we instead used all FIA plots within a bounding box that encompassed a 200 km buffer around both UWCNF and Ashley National Forest, the two national forests in Northern Utah. This enabled a more data-rich analysis that also ideally captured broader regional trends with respect to the subalpine fir's presence/absence. FIA plots were filtered as follows: (1) Only plots measured since 2010 were included; (2) only the most recent plot measurements were included if a plot had multiple measurements; (3) only plots that were sampled were included; (4) only single-condition plots were included; (5) only plots with 0% canopy cover (representing definite "absence" plots) or ≥10% canopy cover (containing full tree measurements to enable a reliable measure of "presence"). Each remaining plot was classified as either "presence" (at least one live subalpine fir tree present within the plot) or "absence". In all, this resulted in a total of 13,791 plots, 750 of which were classified as "presence" and 13,041 of which were classified as "absence".

Another important difference was the suite of predictor variables. All the same terrain and climate variables were used. Landsat 8/9 data were also used but in a slightly different manner. An "early season" image composite was generated in Google Earth Engine, representing the median of cloud-free, snow-free pixels from all images between days 150 and 225 of the years from 2017 to 2021, and a "late-season" composite was generated between days 226 and 300. The goal was to capture some phenological differences in vegetation between the early and late growing seasons. Raw surface reflectance values and all the same spectral indices from Table 2 were used as predictors. Additionally, we incorporated land cover, vegetation structural, and disturbance predictors from the National Land Cover Dataset [104], LANDFIRE [105], the University of Maryland's Global Land Analysis & Discovery lab [106,107], and LandTrendr [59]. Pixel values from each of these predictor layers were extracted at the true (unfuzzed) location of the center subplot for each FIA plot within the study area, the results of which were used in the modeling process.

Rather than the leave-one-out cross-validation approach, we carried out a simple random 70%/30% training/test split to build and assess the accuracy of the model given the greater quantity of available reference data. The accuracy assessment results based on the 30% test data can be observed in Tables A1 and A2. Clearly, the classes were heavily unbalanced (subalpine fir is much less common than it is common throughout landscapes of Northern Utah). Accordingly, measures such as overall accuracy (0.97) are misleading. An example of a more reliable measure of model performance would be F1 for "presence", which balances errors of omission ("presence" is underestimated) and commission ("presence" is overestimated). Our model achieved a "presence" F1 of 0.75, indicating that approximately 3/4 of the variation in "presence" is captured in the resulting model. Some of this uncertainty may be attributable to the scale mismatch between Landsat pixels (900 m$^2$) and the combined area of the four FIA subplots from the National Design (214 m$^2$).

**Table A1.** Confusion matrix for predicting subalpine fir "presence" vs. "absence".

|  |  | Reference Data |  |  |
| --- | --- | --- | --- | --- |
|  |  | Presence | Absence | Total |
|  | Presence | 170 | 54 | 224 |
| Predictions | Absence | 59 | 3855 | 3914 |
|  | Total | 229 | 3909 | 4138 |

**Table A2.** Summary accuracy statistics for predicting subalpine fir "presence" vs. "absence". For class-specific measures (sensitivity, specificity, precision, recall, and F1), "presence" is considered as the "positive" result.

| Metric | Value |
| --- | --- |
| Overall accuracy | 0.97 |
| Kappa | 0.74 |
| Sensitivity | 0.74 |
| Specificity | 0.99 |
| Precision | 0.74 |
| Recall | 0.74 |
| F1 | 0.75 |

**Appendix B. Additional Figures**

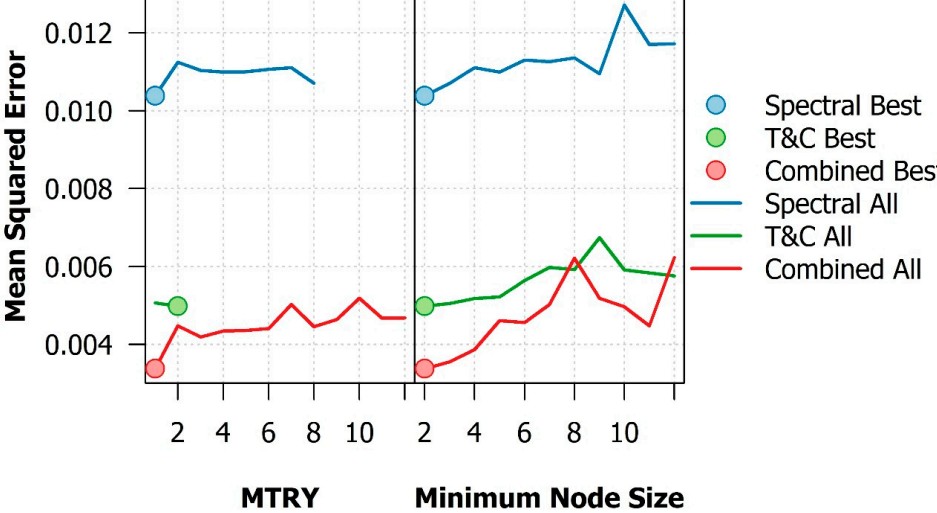

**Figure A1.** Results of the random forest model tuning obtained by using tuneRanger for the two hyperparameters that were tuned in the three infestation predictive models: (1) the number of variables

considered at each tree split (MTRY) and (2) the smallest number of samples that were allowed at the end of each tree node (minimum node size). Lines represent the minima of all combinations tested, and points represent the final parameter selection based on minimizing the mean squared error.

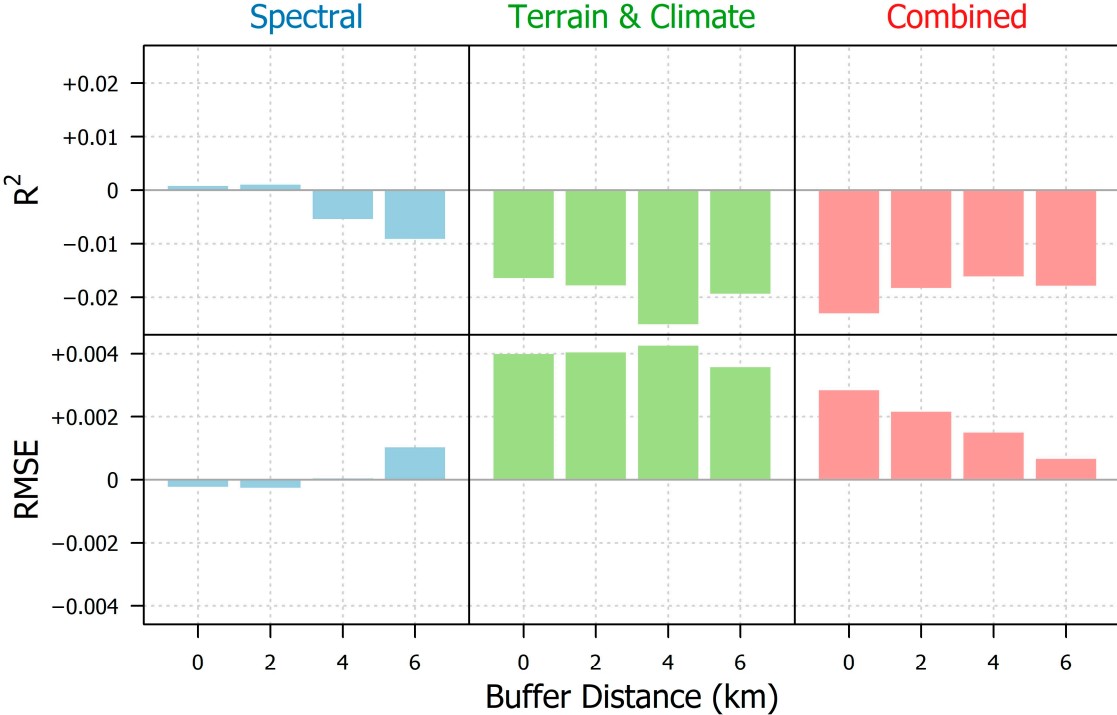

**Figure A2.** Results of a test comparing model performance metrics between a quantile-based bias correction approach and an uncorrected approach. Positive $R^2$ values and negative RMSE values would indicate better performances of the bias correction procedure.

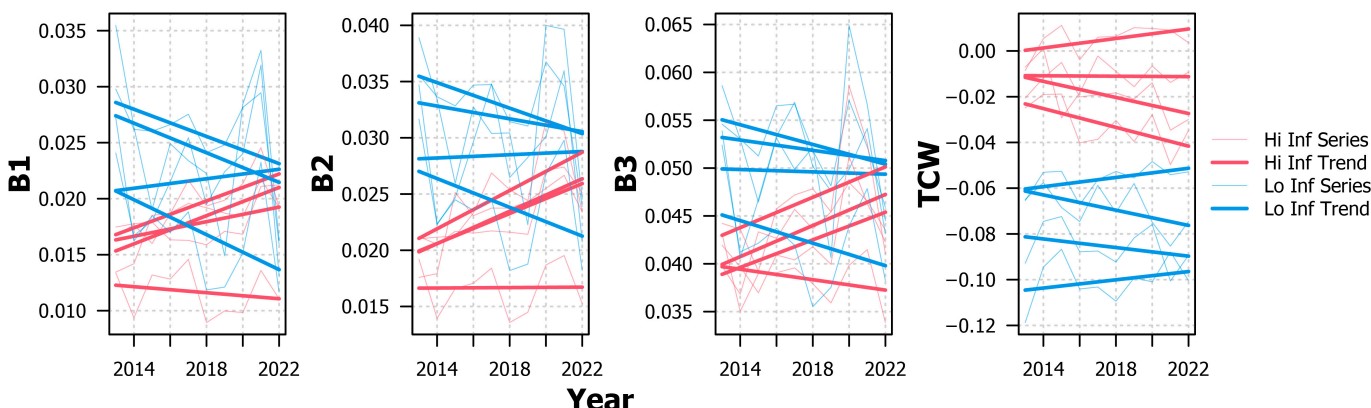

**Figure A3.** Example time series for the four highest and four lowest severity plots in our field database for the four slope-based variables selected in the spectral-only model.

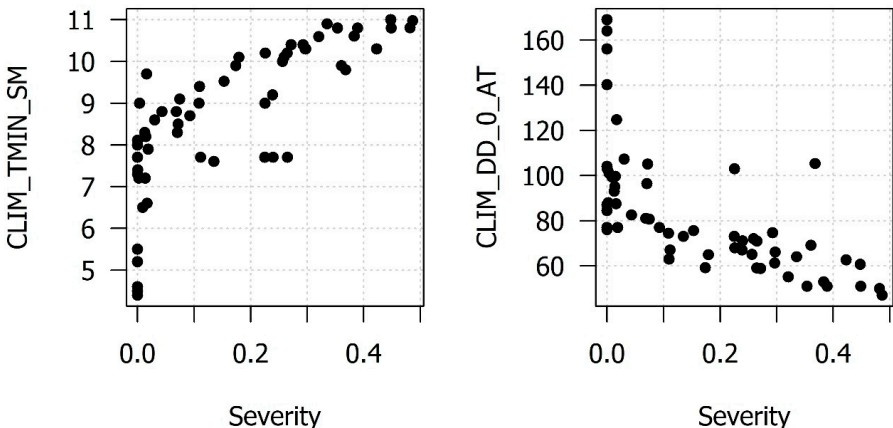

**Figure A4.** Scatterplots comparing plot-level severity to the two final variables selected in the terrain and climate model.

**Figure A5.** Correlation matrix containing all predictor variables used throughout the three BWA infestation severity predictive models. Figure generated using the R package corrplot [108].

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
