# Peer review of "Using Remote Sensing and Climate Data to Map the Extent and Severity of Balsam Woolly Adelgid Infestation in Northern Utah, USA"

_forests, doi:10.3390/f14071357_

Round 1

Reviewer 1 Report

The manuscript is well-written and within the scope of the journal.  I have just few comments to be consider.

The low accuracy results from spectral reflectance data which derived from Landsat sensors (OLI-1 and 2) can be attributed to the moderate spatial (30 m) and multispectral (6 bands only) resolutions. I think using high hyperspectral; high spatial resolution (1 m) images will significantly improve the accuracy of the results. Therefore, authors should be clearly mentioned in the abstract and across the discussion that the remotely-sensed data that used in this study are ‘multispectral’ not ‘spectral’ (e.g. line 17); and the spatial data are ‘moderate-resolution (30 m).  

Reviewer 2 Report

The study has mapped BWA infestation in northern Utah using three guiding principles such as using remote sensing data, terrain and climate data, and combination of both. The article has a relatively clear context and well written. The results are supported by data selection and techniques applied. However, the current form of the manuscript needs some clarity. The specific comments are given below.

Major Comments:

1)      I don’t see any results especially related to objective 4.

2)      In Section 2.2: About 58 plots data was used. The dimension of plots are missing

3)      It has been stated that three machine learning models generated to predict BWA infestation severity. However, which ML methods are used in the article are missing in methodology section. 

moderate edit is required 

Round 2

Reviewer 2 Report

After reading the revised manuscript, I find that the authors have adequately addressed all comments.  

Moderate editing  required